# A high-resolution spatial map of cilia-associated proteins in the human fallopian tube

Feria Hikmet[1], Andreas Digre [1], Jan Niklas Hansen [2,3], Samantha B. Schon[4], Emma Lundberg [2,3,5,6], Matts Olovsson[7], Mathias Uhlén [2,8], Loren Méar [1,9,10] & Cecilia Lindskog [1,10] ✉

Molecular alterations in the fallopian tubes play a pivotal role in the development of cancer and reproductive disorders, yet their molecular landscape at the protein level remains poorly defined. Here, we map key fallopian tube proteins at single-cell resolution utilizing an integrated transcriptomics and proteomics approach. Based on RNA-seq analysis, we identify 310 genes with elevated expression in the fallopian tube, the majority of which are associated with motile cilia function. We spatially characterize 133 of the corresponding proteins in the fallopian tube and other human tissues with motile cilia to subcellular structures of ciliated cells, validating the findings with single-cell RNA-seq and mass-spectrometry data. Eleven proteins previously only studied on the transcript level without information in cilia databases are further analyzed in a hydrosalpinx patient, showing a thinner epithelium, lower density of FOXJ1 expression, and reduced expression of FHAD1, RIIAD1, and C2orf81. Our high-resolution spatial map aids in dissecting the pathways underlying infertility and diseases linked to cilia-specific functions.

Almost one-third of the world's population is affected by disorders associated with reproductive organs, and it has been proven that reproductive health is closely related to overall health[1]. Infertility is increasingly common, and endometrial and ovarian cancer are among the most common cancer types in women. Although recent large-scale efforts have advanced our understanding, key mechanistic aspects of the reproductive system remain unresolved[1–5]. To understand the physiology of the reproductive system and associated pathology, it is necessary to map the proteins that carry out the local functions in each reproductive tissue. The female reproductive system involves multiple organs, with the fallopian tubes (FT) situated as an integral part of the pathway of the gametes during sexual reproduction[6,7]. The architecture of the tissues involved in this process, mainly made up of muscular tubes, inwardly coated with mucosa, is adapted to gently guide the egg and sperm cells towards each other while simultaneously contributing to the natural selection for high-quality sperm cells[8]. Furthermore, after successful fertilization, the early embryo is transported through the FT to the uterus. To facilitate this transport, the FT contains a mucosa with a highly ciliated epithelium interspersed with secretory cells and intercalary cells[9].

Motile cilia are filamentous cellular membrane protrusions that actively move[10]. They are found on the epithelia of several human

[1]Department of Immunology, Genetics and Pathology, Cancer Precision Medicine Research Unit, Uppsala University, Uppsala, Sweden. [2]Science for Life Laboratory, School of Engineering Sciences in Chemistry, Biotechnology and Health, KTH Royal Institute of Technology, Stockholm, Sweden. [3]Department of Bioengineering, Stanford University, Stanford, CA, USA. [4]Division of Reproductive Endocrinology and Infertility, Department of Obstetrics and Gynecology, University of Michigan, Ann Arbor, MI, USA. [5]Department of Pathology, Stanford University, Stanford, CA, USA. [6]Chan Zuckerberg Biohub, San Francisco, San Francisco, CA, USA. [7]Department of Women's and Children's Health, Uppsala University, Uppsala, Sweden. [8]Department of Neuroscience, Karolinska Institutet, Stockholm, Sweden. [9]Department of Women's and Children's Health, Karolinska Institutet, Stockholm, Sweden. [10]These authors contributed equally: Loren Méar, Cecilia Lindskog. ✉e-mail: cecilia.lindskog@igp.uu.se

tissues, including tissues of the reproductive system, the respiratory system, and the walls of brain ventricles[11]. With their movement, motile cilia move entire cells, i.e., the sperm cell, or fluids along epithelial surfaces, such as mucus along the bronchiolar epithelium. The active movement of a motile cilium originates from the axoneme, a central skeleton in the center of the cilium. The axoneme consists of nine outer and, in some motile cilia, also two inner microtubule doublets. Dynein arms interlink the outer microtubule doublets, and their activity leads to tension between the microtubule doublets and the bending of the cilium[12]. Coordinated activation and deactivation of the dyneins result in the active beating of motile cilia[13]. Beyond dynein proteins, the axoneme is rich in additional protein complexes that support ciliary beating, such as radial spoke (RS) proteins. These proteins are required for structural integrity and mechano-regulation of the whip-like movement of the motile cilium[10]. Additionally, motile cilia and sperm also contain signaling proteins and transport proteins to sense and adapt to the environment[14,15]. A diffusion barrier at the ciliary base limits entry and exit of proteins, and the controlled import and export to and from cilia generate a unique protein repertoire in cilia[10]. Genetic diseases resulting in loss of proteins involved in motile cilia function are known to cause primary ciliary dyskinesia (PCD), a disease wherein the movement of the motile cilia is abnormal or entirely lost. The condition is characterized by major symptoms relating to the organs with motile cilia, including infertility[16]. Hydrosalpinx (HS), a pathological distension of the FT due to fluid accumulation, is a common cause of female infertility and an important clinical manifestation of tubal disease[17]. In HS, the normal tubal transport is disrupted, and a mechanical stop for passage occurs, often due to adhesions. This results in impaired ciliary activity in the FT, making HS a clinically relevant model for dysfunction in the FT epithelium. Mapping the genes and proteins involved in motile cilium function is a crucial step toward understanding the molecular mechanisms involved in ciliary motility and the pathophysiology of ciliopathies like PCD[18] or other diseases where motile cilium function is altered, like HS[19].

While the most conserved cilia proteins, such as dyneins and tubulins, have been extensively characterized and multiple protein families related to these proteins have been described in previous studies[10], many genes expressed in ciliated cells lack evidence at the protein level. Spatial proteomics techniques such as classical immunohistochemistry (IHC) allow for detailed localization of proteins within the tissue environment with a resolution that not only reveals in which cell a protein is localized, but in which subcellular region the protein is present, giving clues about the function of the protein.

The spatial expression pattern of substantial portions of the proteome in the different cells of the FT is yet to be explored. To identify key FT-associated proteins and map their localization to different subcellular compartments of cells, we here used an integrated omics approach taking advantage of available transcriptomics and mass spectrometry (MS)-based proteomics data combined with an in-depth immunohistochemistry analysis of human FT, as well as other tissues with motile ciliated cells. We also added testis for comparison with flagella, a structure that shares multiple functional characteristics with motile cilia. Bulk transcriptomics data, derived from the Human Protein Atlas (HPA) resource[20], identified 310 genes with an elevated expression in human FT compared to other tissues and organs in the human body, of which a majority were found to be associated with the structure and function of motile cilia. To characterize these cilia-related genes on the protein level, we applied a stringent IHC workflow with antibodies previously validated as part of the HPA project and mapped the spatial localization of 133 proteins in both FT and other tissues with motile cilia. The effort included localization at subcellular resolution to different compartments of ciliated cells, such as different areas of the cilium structure, or the nucleus and the cytoplasm. Our approach enabled us to identify cilia-related proteins not previously identified on the protein level or described in the context of cilia biology, and investigate their pattern of expression in a tubal pathology context, in HS. This extended map aids in further disentangling the pathways involved in infertility and diseases related to FT and cilia-specific functions.

## Results

### The FT-elevated transcriptome

To assist scientists and clinicians in selecting key proteins for characterizing a specific organ, tissue, or cell type, the HPA project stratifies protein-coding genes into five different categories (tissue enriched, group enriched, tissue enhanced, low tissue specificity, not detected), according to their specificity and distribution of mRNA expression, as previously described[21,22]. The classification system defines how much the expression of a gene is elevated in a particular tissue or cell type compared to the expression levels in the rest of the human body. We leveraged this database to create a set of FT-specific genes (Fig. 1A). We identified 310 genes as most specifically elevated in the FT based on three different categories: 19 tissue enriched genes (at least 4-fold higher expression in FT compared to any other tissue); 114 group enriched genes (at least 4-fold higher expression in a group of 2–5 tissues, including FT, compared to any other tissue); 177 tissue enhanced (at least 4-fold higher expression in a single or group of 2–5 tissues, including FT, compared to the mean expression of any other tissue) (Supplementary Data 1). Most genes were detected (above detection cut-off <1 nTPM) in some or most other tissues, while only two genes were exclusive to FT alone (Supplementary Data 1). Including the FT-enriched genes, 150 of the total 310 FT-elevated genes had the highest expression level in FT (Supplementary Data 2). To provide an overview and explore how the specificity among the group enriched and tissue enhanced portions of the elevated genes are distributed among other tissues with higher expression than FT, we visualized the proportions of tissues with the highest expression (Fig. 1B). The analysis showed that the testis and choroid plexus dominated the number of genes with the highest expression. In contrast, smaller proportions of the genes showed the highest expression in the female tissues, such as endometrium, and the brain, including the basal ganglia and cerebral cortex (Fig. 1B, Supplementary Fig. 1, and Supplementary Fig. 2). In addition to the tissue-based bulk RNA-seq data, we utilized single-cell RNA-seq (scRNA-seq) data from the HPA Single Cell section to determine the cell type specificity of the FT-elevated genes. This dataset showed, as expected, that a large proportion of the FT-elevated genes were elevated in ciliated cells originating from FT, respiratory epithelium, endometrium, or in spermatids, thus identified in cell types harboring motile cilia or flagella. (Supplementary Fig. 1–3). A concatenated Gene Ontology (GO) analysis on the FT elevated genes confirmed that a majority of the FT-enriched genes were associated with motile cilium function and sperm/flagellar motility, with the four biggest GO-term groups being related to either a functional structure of the motile cilium (axoneme and dynein) or cilium movement (Fig. 1C and Supplementary Data 3).

### The spatial localization of FT-elevated proteins in ciliated and non-ciliated cells

Next, to spatially profile the FT elevated gene products, we used the stringent validation pipeline adopted within the HPA workflow. This workflow validates the staining patterns of antibodies in an immunohistochemical setting and provides the best estimate of true protein expression levels[23]. Of the 310 genes, 72% (223 genes) of the corresponding proteins had published IHC data from antibody-based protein profiling, with reliability scores as previously defined by the HPA[23]. Expression profiles of 59 proteins rated as Uncertain were omitted from the analysis due to inconclusive antibody reliability, resulting in 164 potential protein candidates. All candidates were revisited and manually evaluated, omitting an additional 31 candidates due to poor

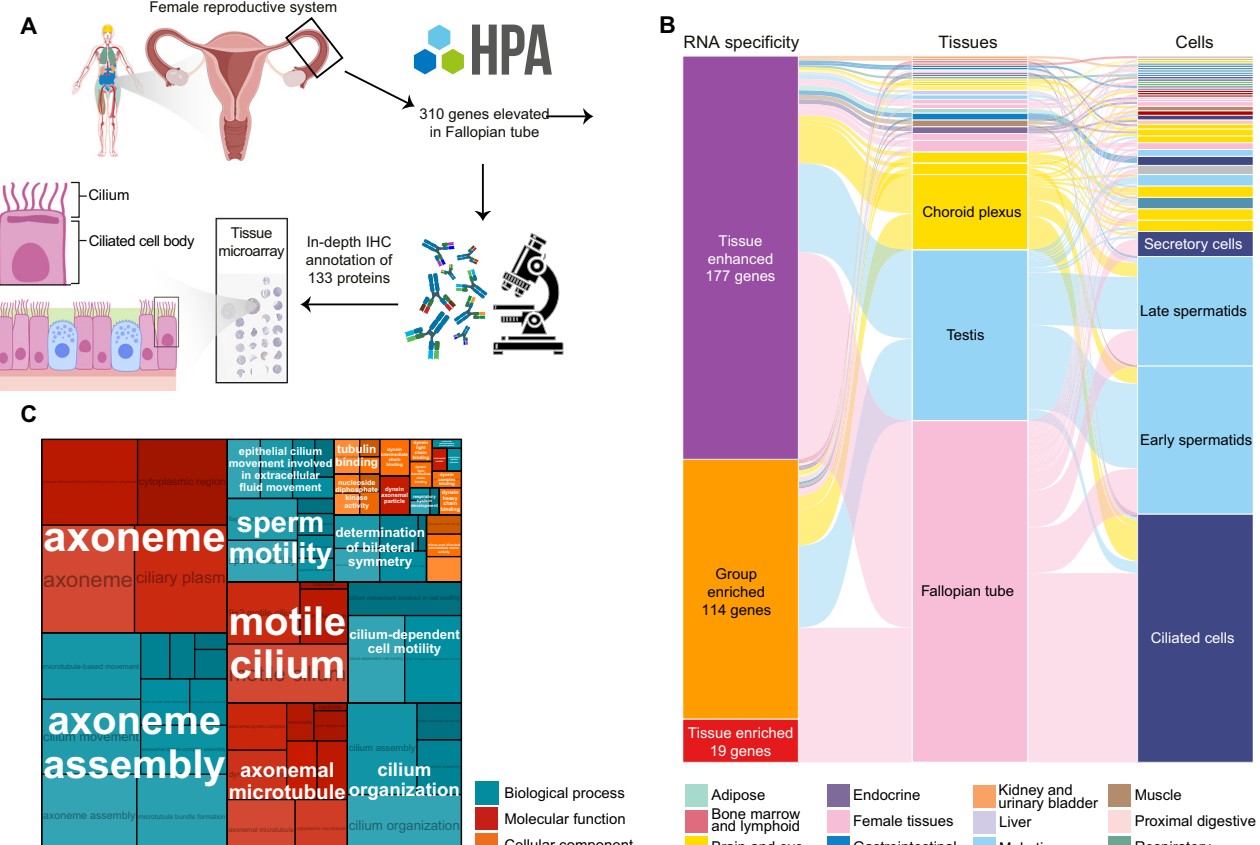

**Fig. 1 | The expression, functions, and characteristics of the FT-elevated transcriptome. A** Bulk transcriptomics data of FT tissue was used to select genes with different specificity to FT based on RNA categories retrieved from the HPA. In total, 310 genes were elevated in FT, and antibodies targeting 133 proteins were identified for further analysis in the current study based on stringent antibody validation criteria. Created in BioRender. Hikmet Noraddin, F. (2026) https://BioRender.com/k3x2119. **B** RNA specificity for the 310 genes elevated in FT, and the tissue and cell types with the highest expression of each gene are presented in an alluvial plot (**C**). GO enrichment analysis for all FT-elevated genes. GO term enrichment was assessed using an over-representation test with Benjamini–Hochberg correction for multiple testing. Terms with adjusted *p*-values < 0.01 were considered significant. Source data are provided as a Source Data file.

signal-to-noise ratio or too low antibody volume. In the final list of 133 proteins selected for in-depth IHC profiling (Fig. 1A), 109 had evidence at the protein level, 21 proteins had evidence at the transcript level, and three proteins lacked evidence as defined by UniProt[24]. The evidence levels of the 133 proteins were also cross-referenced with the number of available GO terms, indicating that multiple proteins lack GO terms, suggesting limited information on the potential function[25] (Fig. 2A and Supplementary Data 4). Antibody profiling allowed us to link specific proteins to ciliated or non-ciliated FT epithelium (FTE) phenotypes (Fig. 2B) and revealed the subcellular localization of multi-protein complexes involved in cilium motility, biogenesis, and maintenance. The IHC staining of FTE showed that 123 proteins were localized exclusively in ciliated FTE cells, 7 in non-ciliated cells, and three proteins (FBXO21, MUC16, and PTGIS) showed positivity in both cell types (Fig. 2C). Next, we determined the overlap of our ciliated cell-exclusive IHC dataset (*n* = 123) with databases and resources suited for exploring potential cilia-related proteins across species (Fig. 2D). These datasets constitute both manually annotated gene lists or compiled with statistical models and bioinformatics at a systems-level: The CiliaCarta compendium is based on a statistical integration of datasets to predict ciliary genes[26]. The SysCilia V1 and V2 consist of curated lists of proteins related to ciliogenesis, found in cilia, or associated processes[27,28]. The Nevers dataset stems from a phylogenetic profiling of ciliated and non-ciliated species[29]. The Karunakaran list is based on bioinformatically predicting interaction partners for known ciliary genes[30] (Supplementary Fig. 4A). By merging the cilia databases for easy overview, a

large portion of the proteins were included in at least one of the cilia databases (Fig. 2D). Interestingly, 34 proteins included in our study were not identified in any of the databases. Three proteins (CFAP206, C1orf87, and NME9) were only detected in the bioinformatic and phylogenetic datasets. Thus, our data confirms the bioinformatic and phylogenetic predictions (Supplementary Fig. 4A).

We next compared the 123 proteins that we characterized with IHC in ciliated cells with datasets of sperm and flagellar proteins acquired with MS (Supplementary Fig. 4B), comprising 14 studies on the proteome of human sperm flagella[31] including one study on changes in protein phosphorylation related to motility of human sperm[32], and one study on the flagella of sea urchins, sea anemones, and choanoflagellates, providing insight into conserveness of the proteins that we identified[33]. More than half of the 123 proteins (*n* = 70) we characterized appeared to be highly conserved across species since they were also detected by the work of Sigg et al.[33]. Two-thirds of the 123 proteins (*n* = 85) were detected in sperm flagella with MS, and interestingly, there were some genes (*n* = 14) only detected in sperm but not in the cilia databases (Fig. 2D). Our study suggests adding them to cilia databases.

Lastly, we examined the association of cilia-associated proteins with ciliopathy-related genes. Ciliopathies are genetic diseases that have been linked to the dysfunction of cilia. Many ciliopathies related to genes specifically localizing to motile cilia comprise infertility phenotypes. We compared our list of proteins to the ciliopathy database CiliaMiner by the Kaplan lab (https://kaplanlab.shinyapps.io/

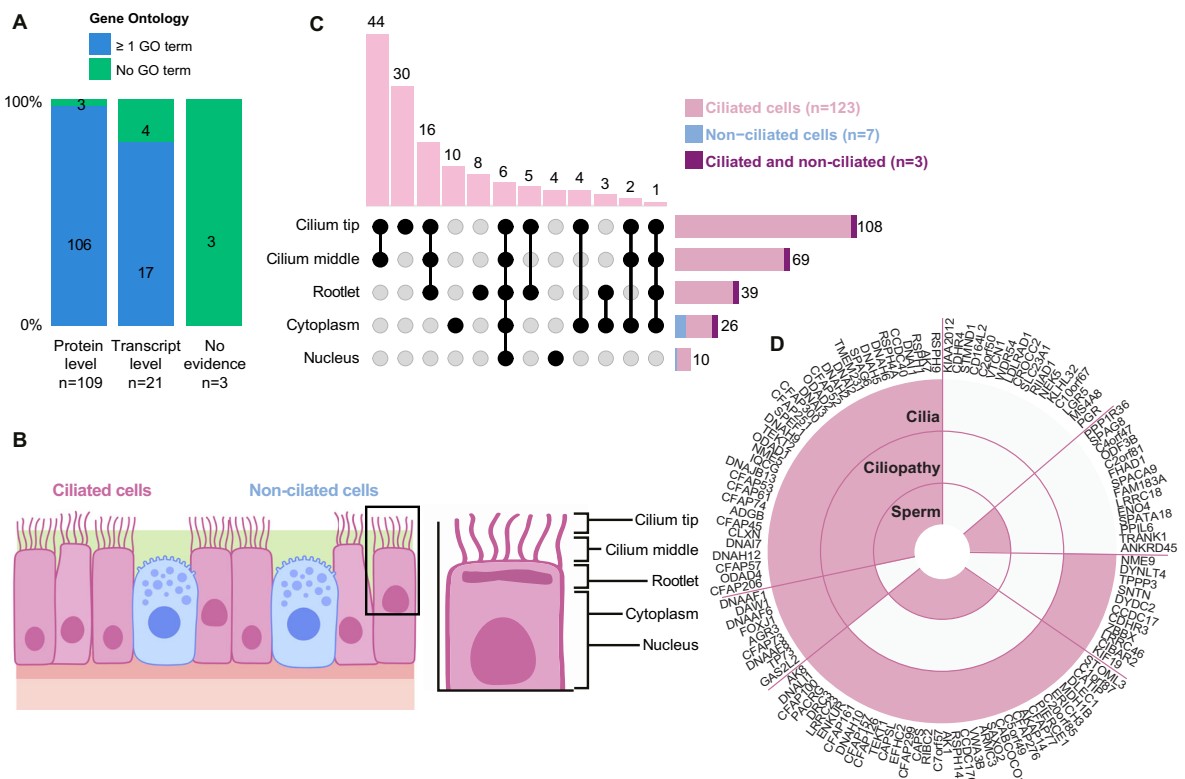

**Fig. 2 | FT-elevated genes are mainly expressed in ciliated FT epithelial (FTE) cells. A** Stacked percentage bar chart to highlight the level of protein characterization of the proteins that, based on IHC were expressed in ciliated cells, using UniProt protein evidence, as well as available Gene Ontology terms. **B** Schematic image showing the cell types and subcellular compartments that were annotated in ciliated FTE cells. **C** UpSet plot showing the frequency and distribution of the staining in FT mucosa at the subcellular level. The schematic image illustrates the various parts of ciliated cells that were annotated, along with the frequency of each annotation combination. Portions of positive stainings in ciliated cells (pink), non-ciliated cells (blue), and both (purple) are shown. **D** The 123 cilia proteins were examined in IHC in databases for cilia-relevant function and expression (cilia), a database for ciliopathies called CiliaMiner (Ciliopathy), as well as comparing against flagellar proteins identified in different studies (Sperm). Source data are provided as a Source Data file.

ciliaminer)[34]. Only 27 of the 123 genes were listed as ciliopathy-associated (Fig. 2D and Supplementary Fig. 4C). An additional 17 genes were listed as potentially ciliopathy-related. Among the listed ciliopathy-related genes, approximately half of them were related to motile ciliopathies, validating their role in ciliopathy-related motility.

A graphical summary of the 123 proteins experimentally characterized with IHC in this study in the different compartments of ciliated FTE cells is presented in Fig. 3. The schematic representation overviewed the current level of protein evidence and cilia knowledge for these proteins and was grouped into characterization groups (Groups 1–7) to highlight poorly defined proteins to high levels of characterization, as defined by the existing information in biological knowledge resources and databases (Fig. 3). Representative IHC images of the FTE for these proteins are shown in Fig. 4.

**Spatial analysis of cilia-associated proteins in other tissues with motile cilia**

The FT expresses many proteins that are also expressed in other tissues, especially in the testis, where sperm are produced, which employ a motile cilium, the flagellum, for motility[35]. Since most tissue-elevated proteins in FT were associated with the motile cilium, we next expanded our antibody-based analysis to other tissues that shared gene expression specificity with FT and were known to harbor motile cilia, namely endometrium, cervix, nasal mucosa, and bronchus, the epithelial ventricle lining close to caudate, choroid plexus, and epididymis. Additionally, we also included samples of testis to compare the expression and localization between the cilium and the flagellum. Therefore, we assembled a tissue microarray (TMA) including samples

of FT, proliferative and secretory-phase endometrium, endocervix, nasal and bronchial epithelium, ventricle ependymal epithelium, and choroid plexus epithelium in the brain, as well as male reproductive epididymis (efferent ducts) and testis. The TMA was next stained with all 133 antibodies included in the study to determine their spatial protein expression in the other ciliated tissues. Of the 123 motile-cilium-associated proteins, 51 were present in all tissues, and 44 in all except the choroid plexus, indicating a shared core set of proteins for motile cilia across tissues (Fig. 5A). The epithelial cells in the choroid plexus form 9 + 0-type motile cilia during development that are degraded around birth[36]. In agreement, we saw no clear staining of cilia among choroid plexus epithelial cells, except for four proteins localized to the cilium basal area, supporting this recent study on murine choroid plexus[36]. The positive staining in choroid plexus for 64 proteins was mainly localized to the cytoplasm (Fig. 5A and Supplementary Data 5). Representative IHC stainings of proteins spanning groups 1 to 7 across different tissues are shown in Fig. 5B to showcase the diversity of localization across tissues.

The ten proteins that were localized to non-ciliated FTE cells were also characterized in the other tissues. These proteins showed a variable staining pattern across tissues, some with positivity in ciliated cell structures, but with additional staining in the cytoplasm and nucleus of non-ciliated cells. None of these candidates appeared to be cilia-specific after validation in the other tissues (Supplementary Fig. 5).

**Validation of FTE cell specificity**

Next, we aimed to validate our IHC dataset in FTE cells. To this end, we applied a two-part validation approach where we compared our

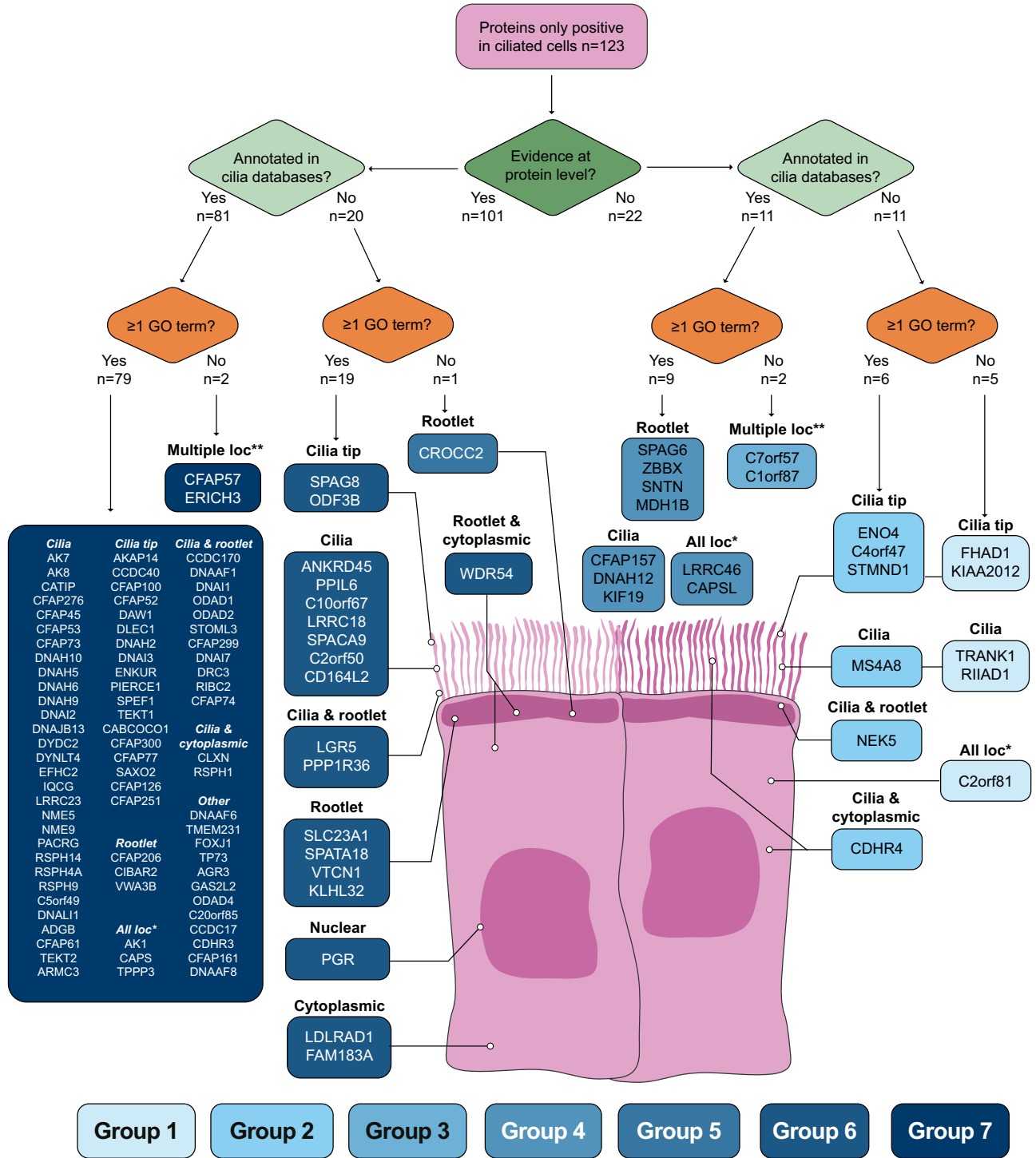

**Fig. 3 | Overview schematic illustrating the 123 proteins stained exclusively in ciliated cells of the FTE.** The proteins are separated at different levels to give an overview of the current knowledge and status of biological characterization with UniProt, Cilia databases, and GO terms. Each protein and its staining in each compartment are highlighted. The groups represent proteins with low (light blue) to high (dark blue) levels of characterization. Source data are provided as a Source Data file.

protein expression data with reanalysis of scRNA-seq and MS datasets from human FT tissue. For the scRNA-seq analysis, we integrated three previously published FT scRNA-seq datasets[3,5,37]. Clustering the data and identifying cell types by established cell markers made it possible to identify the most common cell types that constitute the FT tissue, namely ciliated cells, secretory cells, fibroblasts, smooth muscle cells, endothelial cells, and different types of immune cells (Supplementary Fig. 6). It also facilitated the

comparison of IHC data with gene expression on the single-cell level, however, the cell types were merged for practicality and simpler validation.

As a further step of validation, we utilized an existing spatial single-cell proteomic dataset of the human FT generated using Deep Visual Proteomics (DVP), a single-cell MS platform[38]. FOXJ1-positive and negative cells were antibody-labeled, microdissected, and analyzed by high-sensitivity MS. This provided independent support for

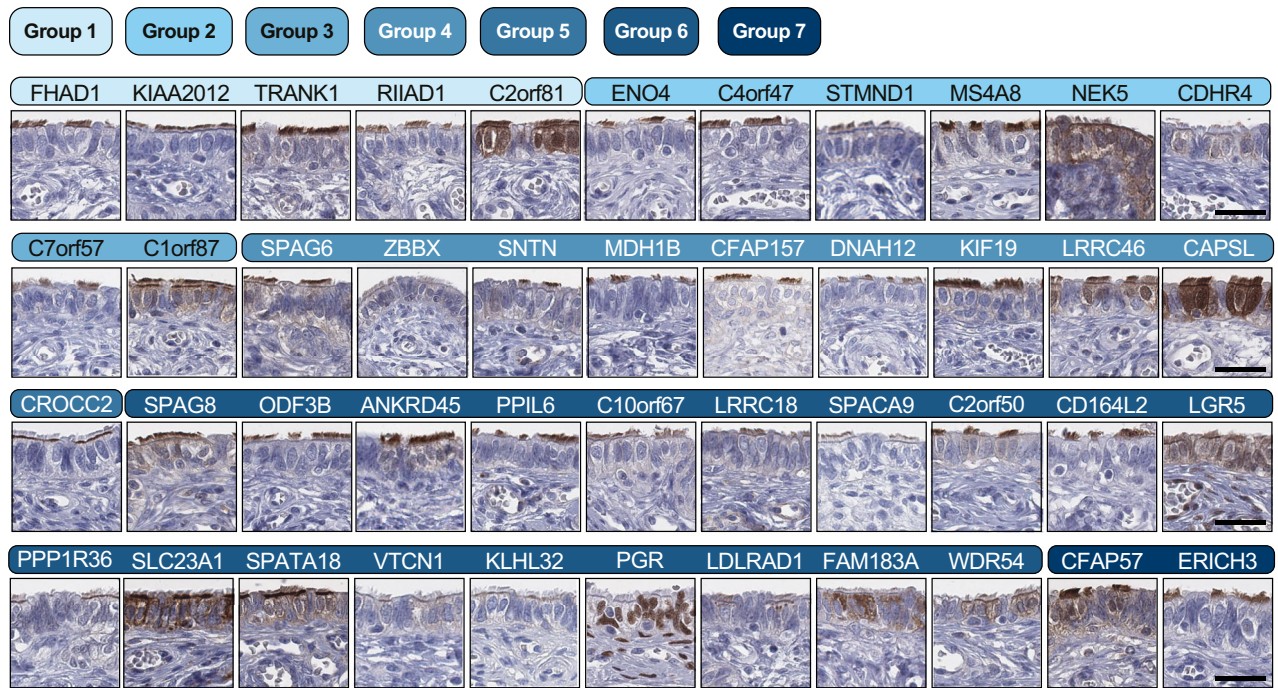

**Fig. 4 | Representative IHC images of proteins stained in ciliated FTE cells in FT with varied levels of evidence and association with cilia function.** The colored bars represent their characterization group, ranging from low (light blue) to high (dark blue) level of characterization, derived from Fig. 3, with the protein names above each IHC image. The stainings were performed once per protein target (see Methods for the number of included tissues in the TMA). Scale bars, 50 μm. Scale bars are representative of all images in the figure. Source data are provided as a Source Data file.

our findings. Among the 133 proteins analyzed with IHC, 95 proteins were included in this MS dataset.

In general, the proteins were consistent between the datasets, and, e.g., the ciliated-cell-specific proteins showed a high expression level in the ciliated cell scRNA-seq cluster. By applying row-wise hierarchical clustering on the scRNA-seq expression levels, the two upper clusters showed a congruent match between the three datasets, highlighting the most abundant ciliated-specific gene based on scRNA-seq levels (Fig. 6A). A consistent pattern of congruent expression between the datasets was also observed for the largest third cluster. However, two genes showed a mismatch between datasets. The progesterone receptor (PGR) was expressed in both ciliated and non-ciliated FTE cells according to scRNA-seq, while IHC only showed staining in ciliated FTE (Fig. 6A, B). Additionally, alternative antibodies targeting different antigenic sequences and validated in the HPA resource confirm the staining pattern in ciliated cells in the FTE, similar to the PGR antibody used in this study (Supplementary Fig. 7). Furthermore, Stomatin Like 3 (STOML3) was, according to scRNA-seq and IHC, specifically expressed in ciliated cells. However, the MS data pointed to non-ciliated cells, mostly being enriched in FOXJ1-negative cells (Fig. 6A). Among the genes in the fourth cluster that displayed varied scRNA-seq expression, four proteins showed differences between the validation datasets and IHC. For MUC16, IHC showed a broad staining pattern with both ciliated and non-ciliated cell types stained, but scRNA-seq data mainly showed high expression in the non-ciliated cell type, and the MS-based data pointed to ciliated cell-specificity (Fig. 6B). Similar to PGR, alternative antibodies against MUC16 were available on the HPA resource. Interestingly, one of them was specific for non-ciliated cells, while the other antibody showed positive stainings in all luminal membranes of FTE cells, partly validating our results (Fig. 6B and Supplementary Fig. 7). VTCN1 lacked MS-data, but expression in scRNA-seq was mainly in non-ciliated cells, while the IHC clearly labeled ciliated FTE cells (Fig. 6B). Interestingly, *PTGIS* scRNA-seq expression was mainly detected in the other cell type cluster. PTGIS IHC showed a broad staining in most cells, but the MS

data marked PTGIS as enriched in FOXJ1-negative cells. An alternative anti-PTGIS antibody from the HPA also showed the same staining pattern as the antibody included in this study, albeit at lower intensity (Supplementary Fig. 7). For SLC27A6, scRNA and IHC were congruent in their specificity for non-ciliated cells, while the MS data pointed at ciliated cell-specificity (Fig. 6B). FBXO21, present in the lower heatmap cluster, was indicated by scRNA and MS to be a non-ciliated gene and protein. However, IHC showed immunostaining in both cell types (Fig. 6B). For both LGR5 and TRANK1, scRNA-seq data showed an association with non-ciliated cells, while IHC stained the cell body and cilia in the two proteins, respectively. No MS data was available for these two proteins.

## HS disrupts fallopian tube epithelial organization and protein expression

To explore our top candidate proteins in a clinically relevant disease model with known disruption of the FTE, we used IHC to assess staining intensity and quantity in HS, a FT pathology caused by chronic obstruction and inflammation[39]. We first assessed protein expression patterns by IHC across three FT controls and one HS sample (Fig. 7A and Supplementary Data 6). The stained slides were scored by manual annotation, and FOXJ1 staining was also included as a technical control staining for monitoring the ciliated cell population in the FTE (Fig. 7B). Eleven candidate proteins from Groups 1 and 2 were selected, and most proteins in the two groups showed no difference in staining in controls and HS. However, three proteins (FHAD1, RIIAD1, and C2orf81 showed a marked reduction in the number of stained cells and lower staining intensity in HS (Fig. 7C). Histopathological studies have shown that HS tissue undergoes extensive epithelial remodeling, including flattening of the mucosa, loss of ciliation, and chronic inflammatory changes. Therefore, we were interested in evaluating the epithelial mucosa architecture and width (or thickness), and ciliated cell density in our HS sample compared to healthy controls. Because FOXJ1 is a lineage-defining transcription factor required for motile ciliogenesis[40], its expression provides a clear readout of ciliated cell

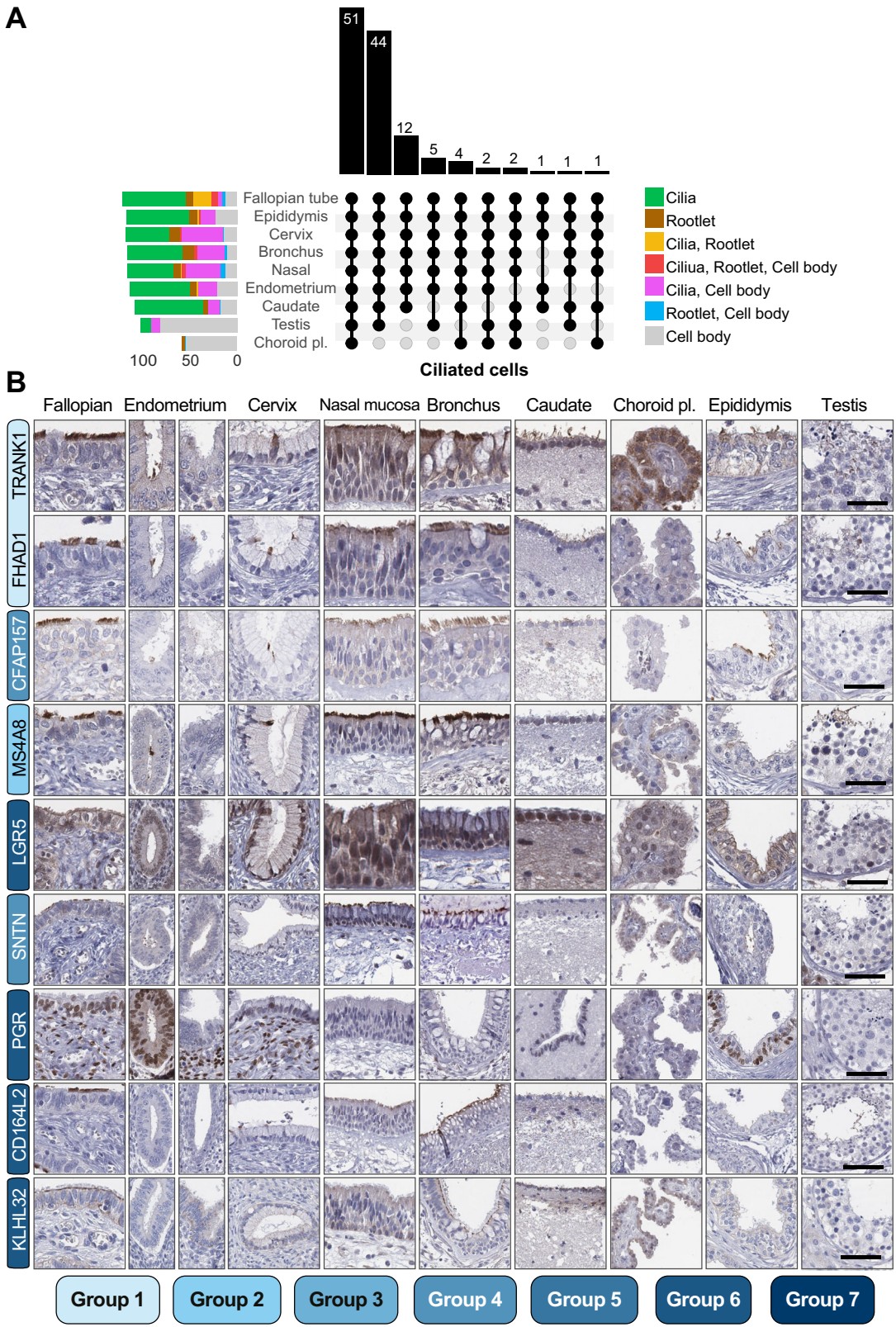

abundance in FTE. We analyzed HS tissue alongside the control FT tissues to test whether the disruption in HS pathology is characterized by altered FOXJ1 expression and reduced ciliated cell density. In Fig. 7D, the analysis is exemplified by control sample C1 (FOXJ1 IHC staining and epithelial width measurement for all samples are presented in Supplementary Fig. 8). For control FTE, FOXJ1-positive nuclei were abundant, whereas HS displayed visibly reduced labeling (Supplementary Fig. 8). Automated image analysis of annotated epithelial regions revealed a significant reduction in epithelial width in HS compared with controls (Fig. 7E), and when normalized to epithelial width, both the density of FOXJ1-positive nuclei and staining areas were lower in the HS sample, indicating a depletion of ciliated cells.

**Fig. 5 | The expression of FT-elevated proteins in human tissues with motile ciliated epithelial cells. A** UpSet plot of protein expression in tissues with cilia-covered epithelium of those expressed in ciliated FTE cells. The bar chart at the top shows the number of proteins associated with each unique combination of sub-cellular localizations indicated below. The colored bars on the left-hand side represent the subcellular annotations derived from each tissue. Note that proteins only stained in ciliated FTE cells are included. Please note that the Cilia annotation also includes flagella localization for proteins annotated in the testis, and the Cell body includes cytoplasmic, membranous, and/or nuclear localization.
**B** Representative IHC images from each upright column in (**A**) to showcase proteins

of all staining patterns, including stainings of TRANK1, FHAD1, CFAP157, MS4A8, LGR5, SNTN, PGR, CD164L2, and KLHL32 in FT, proliferative (left) and secretory (right) phase endometrium, endocervix, nasal and bronchial epithelium, caudate ventricle walls and choroid plexus epithelium in the brain, as well as male reproductive epididymis (efferent ducts) and testis. The colored bars for each protein name represent their characterization group, ranging from low (light blue) to high (dark blue) level of characterization, derived from Fig. 3. Scale bars, 50 μm. Scale bars are representative of all images in the figure. Source data are provided as a Source Data file.

## Discussion

Here, we explored the FT-specific proteome at the tissue, cellular, and subcellular levels with spatial resolution using antibody-based imaging. Based on bulk RNA-sequencing, we identified 310 genes that were elevated in FT compared with other human tissues. We mapped the protein expression of 133 of those FT-elevated genes, out of which a majority ($n = 123$) were expressed in ciliated cells. The analysis was performed with a level of resolution that enabled us to characterize the distribution of proteins within the subcellular compartments of epithelial cells in FT and other organs carrying out functions where motile cilia play an important role.

Based on an integrated approach combining IHC and scRNA-seq with functional validation utilizing multiple publicly available datasets, including MS, we here performed a thorough characterization of proteins expressed in ciliated cells. Out of 133 proteins with reliable antibodies for detection, a majority were localized exclusively to ciliated cells ($n = 123$), and some to non-ciliated FTE cells. Three proteins were mapped to both cell types. The ciliated cell-specific proteins were validated with both scRNA-seq and MS datasets, with no substantial disagreement with protein cell type-specificity measures from either of the two orthogonal methods. Only seven proteins showed higher expression in other cell types than ciliated cells in the scRNA-seq dataset, disagreeing with our IHC profiling, of which only three proteins (PTGIS, LGR5, and TRANK1) were clearly more specific to other cell types, while the other four proteins (FBXO21, PGR, MUC16, and VTCN1) had comparative expression in multiple cell types. Why the data differ for these proteins remains unclear and can result from a number of factors, such as differential regulation of RNA and protein for these genes or variable expression levels across patients. Regarding the seven proteins mapped exclusively to secretory cells, only OVGP1 lacks evidence at the protein level according to UniProt. All seven proteins were validated as secretory cell-specific in the scRNA-seq data. In summary, we provide a robust and highly validated dataset.

Among the 123 proteins that we identified within ciliated FTE cells, 22 proteins previously lacked evidence at the protein level according to UniProt, and many more proteins have not previously been shown to localize to ciliated cells and the cilium with this level of detail. For comparison, the MS study[38] only included data for two proteins that previously lacked evidence at the protein level: MSLN, which showed significant specificity towards FOXJ1-negative non-ciliated cells, and SLC27A6, which lacked significant association with either FTE cell type. This highlights the strength of combining the different analysis approaches and the complementary value of our IHC-based dataset. Similarly, a considerable number of the proteins analyzed in this study were not included in cilium databases that list genes and proteins across species related or localized to cilia based on different types of evidence. As many as 31 of the 133 proteins studied with IHC were not included in any of the cilium databases CiliaCarta, SysCilia, Nevers, and Karunakaran[26,27,29,30], making the data of this study a valuable contribution to the efforts towards mapping all cilia genes and proteins.

The well-characterized protein CFAP45 has been shown to be associated with three distinct types of ciliopathies, highlighting its broad relevance to motile ciliary function[41]. In agreement, we detected the protein CFAP45 in motile cilia of all studied tissues except in the

choroid plexus. This is consistent with recent findings[42], where it was demonstrated that CFAP45 mutations cause asthenospermia due to dyskinetic sperm flagella, establishing its role in male fertility. Although affected individuals exhibit only mild respiratory symptoms and do not meet diagnostic criteria for primary ciliary dyskinesia (PCD), the reproductive phenotype is prominent and clinically significant. This underscores the importance of CFAP45 in sperm motility and suggests its inclusion in the diagnostic evaluation of motile ciliopathies with reproductive involvement, for which our study has highlighted a well-validated antibody that could be used in diagnostic applications.

Mouse and Swine studies have shown that the cilia of choroid plexus epithelium undergo significant remodeling during the prenatal development and early period of life, being remodeled around birth and showing $9 + 0$ axoneme arrangements[36]. In line with this, we did not detect proteins in choroid plexus cilia that are part of the central microtubule pair (SPAG6) or radial spokes (RSPH1, RSPH4A, RSPH9, RSPH14, NME5), which link the central microtubule pair to the outer microtubules, while we identified them in other motile cilia within the same experiment. In our bulk RNA-seq dataset, we observed that the choroid plexus expresses other genes related to ciliary motility, such as dynein arms. While we were not able to identify a similar type of expression in motile cilia as shown in other tissues, we still could see many of them in the cytoplasm. This poses the important question of whether choroid plexus cilia in human adults are, in fact, motile, but the proteins may still be expressed. A recent study has demonstrated, mostly in murine tissues, that choroid plexus cilia are degraded postnatally[36,43]. This could explain why we failed to detect motile cilia-related proteins in the choroid plexus in adult human samples. This observation also sheds light on the fact that the distinction between motile and immotile (so-called primary) cilia might be less sharp than anticipated over the last few decades. There could be a continuum between primary and motile cilia, and future research should be directed towards addressing this question[44]. Confirming this hypothesis, primary cilia have mostly been ascribed to a sensory antenna role in tissues, although their function and the underlying mechanisms have not yet been fully revealed. We recently showed that kidney primary cilia contain motile cilia proteins[45] and mouse pancreas islets reportedly have primary cilia featuring motile cilia proteins[46]. Conversely, there is evidence that motile cilia might also serve as antennas. For example, airway epithelia reportedly contain bitter-taste receptors that trigger an increased cilium beating frequency when exposed to bitter substances[47]. This mode of action was postulated to be a defensive mechanism where noxious substances in the airways can initiate the removal of potentially dangerous compounds. Sperm flagella also feature many signaling proteins and receptors, and thus, are sensors and motors at the same time[48]. In fact, we also discovered a few proteins in cilia related to calcium signaling and calcium channels (CABCOCO1, CAPS, CAPSL, EFCAB1, ENKUR), and cAMP signaling (AKAP14). Yet, the role of motile cilia as sensors remains underexplored.

Most of the FT-elevated proteins were also detected in the testis. However, those that were negative were mainly related to basal body organization and epithelial differentiation. Several of these, including

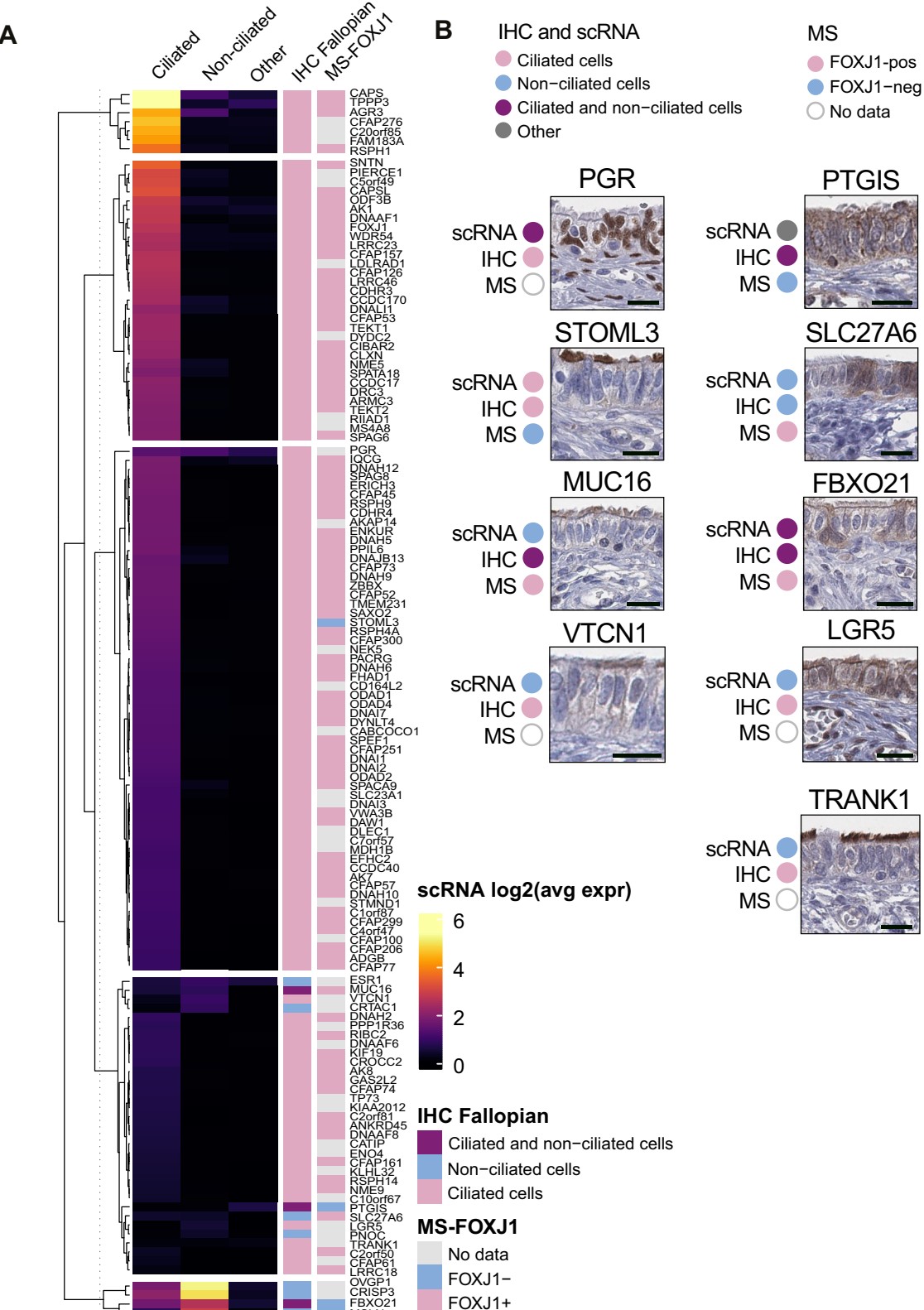

**Fig. 6 | Validation of IHC profiles by orthogonal methods and cilia-related databases. A** Validation of IHC annotation data for 133 proteins from the FT elevated genes using two independent methods. The panel shows a heatmap of log2 expression levels from integrated scRNA-seq data across ciliated cells, non-ciliated epithelial cells, and other cell types. Rows were clustered using a K-means algorithm (k = 5). The annotated FTE cell specificity for each protein is shown in the left annotation bar, and ultra-sensitive MS data from laser micro-dissected FOXJ1-positive and FOXJ1-negative human FTE cells are displayed in the right annotation bar. **B** Highlighted proteins with partial matches across the three datasets, including a cell-specificity summary for each dataset next to the corresponding image. The results for each dataset are shown below the corresponding protein name. Scale bars, 20 µm. Source data are provided as a Source Data file.

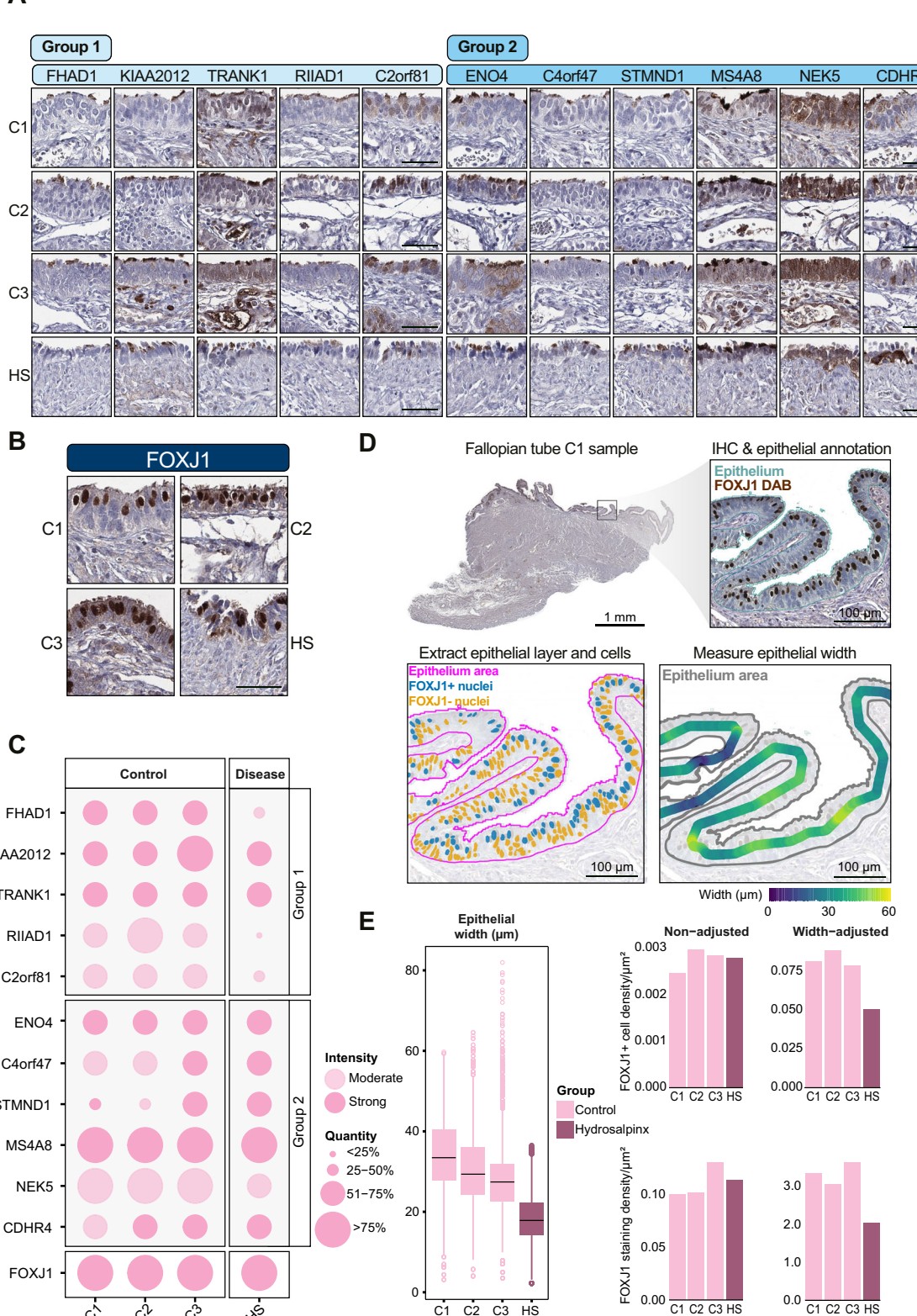

**Fig. 7 | Loss of FOXJ1⁺ ciliated cells and altered candidate protein expression in a hydrosalpinx patient. A** Representative IHC of Groups 1 and 2 candidate proteins and **B** FOXJ1 staining in FT ampulla from three controls (C1, C2, and C3) and one hydrosalpinx (HS) sample. Scale bars, 50 μm. Scale bars in (**A**) are representative of all images in (**A**). Scale bars, 50 μm (**A** and **B**). **C** Dot plot summarizing IHC staining intensity and frequency for candidate proteins from groups 1 and 2.

**D** Representative IHC images for group 1 and 2 candidates in controls and hydrosalpinx. **E** Quantification of epithelial width and density of FOXJ1-positive cells. Boxplots show the median (center line) and interquartile range (IQR; box), with whiskers indicating the data spread beyond the quartiles. Outliers are shown as individual circles/dots. Source data are provided as a Source Data file.

CIBAR2, GAS2L2, CROCC2, and TPPP3, are involved in basal body anchoring, rootlet formation, and microtubule stabilization. These functions are important for the organization of multiple motile cilia in epithelial cells but might not be required for the formation of a single flagellum in sperm. In addition, PGR, ESR1, OVGP1, and AGR3 are markers of epithelial differentiation and hormonal regulation in the FT. The lack of these proteins in the testis likely reflects the different cellular environment, where ciliogenesis occurs in germ cells without epithelial polarity or hormone responsiveness. These differences highlight how the shared axonemal core proteome of cilia and flagella is complemented by tissue-specific modules, allowing to explore ciliary specializations and to better understand phenotypic variability in ciliopathies[49].

To harmonize the data and enable a robust validation strategy, we limited our analysis to expression in ciliated and non-ciliated cells. Intercalary cells, or peg cells, constitute an important cell type in the FTE, as these cells have regenerative features[50]. It has also been stipulated that these cells might be precursors of cancerous cells that develop serous ovarian cancers[50,51]. Since none of the 133 proteins analyzed with IHC displayed specific localization to intercalary cells, these cells were not separated as an independent group. The absence of intercalary-specific proteins may reflect their transitional phenotype, low abundance, or limited molecular distinction from neighboring cell types. Another limiting factor in this study is the low number of replicates used for each tissue, and it is possible that some of the tissues comprising the TMA originate from patients with genetic ciliary dysfunction, a clinicopathological parameter that is not possible to retrieve in the present tissue collection. Also, IHC tissue analysis for determining the exact localization of cilia proteins might be subject to variable accuracy for different proteins. For example, low-abundant proteins in small structures may be difficult to detect or observe in IHC images. Also, highly multi-localizing proteins across multiple subcellular structures may be difficult to distinguish into the specific regions or to be detected in the regions where abundance is lower than in other regions. Lastly, it is highly relevant to investigate whether hormonal status could affect the expression of the proteins identified in this study. Menopause and other hormonal transitions could reshape FT biology[52]. Future work integrating these resources could reveal how endocrine changes modulate key protein pathways, offering deeper insight into tissue remodeling and disease susceptibility. Nevertheless, for a majority of the proteins analyzed by IHC, the staining pattern was consistent between samples, and we believe that the generated data constitute a fair estimate of the protein expression in ciliated cells.

The DVP workflow provides a spatial and robust method with high sensitivity and specificity[38]. The DVP approach provided at best a cellular resolution, whereas our IHC-based approach allows subcellular protein identification. By labeling ciliated cells and distinguishing them from secretory cells with the cell lineage-specific transcription factor FOXJ1, thousands of proteins could be measured with the unbiased DVP approach in the two different groups. The transcription factor FOXJ1 is essential for the formation of motile cilia throughout the animal kingdom. Target genes, therefore, likely constitute an important part of the motile cilia program. However, since cell selection was based solely on a nuclear marker, proteins located in distal structures such as cilia may have been excluded from the laser-microdissection, which could be the source of some of the disagreement between the DVP and the IHC data. Therefore, detailed spatial analysis with conventional methods, such as IHC, provides important complements to DVP analysis, generating a high-resolution map of individual proteins[53].

The epithelial mucosa in the analyzed HS sample was flatter than the control tissues, recapitulating previous observations[19]. Protein expression analysis in HS of unknown proteins from Groups 1 and 2 effectively showed that the expression of FHAD1, RIIAD1, and C2orf81

was lower in the HS individual compared to the controls, where the expression was stable. Interestingly, the spatial expression pattern for these three proteins was confined to the motile cilia of FTE. As stated previously, C2orf81 remains largely uncharacterized, with limited functional annotation available. However, the forkhead-associated domain-containing protein 1 (FHAD1) is a large multi-domain protein, and FHAD1 null mice are reported to show reduced sperm motility and meiotic defects[54]. Comparably, RIIa domain-containing protein 1 (RIIAD1) is much smaller and relatively uncharacterized. The presence of an RIIa domain, a modular protein–protein interaction motif also found in several adenylate kinases (AKs), is particularly intriguing. In motile cilia, AKs use this domain to anchor to radial spokes, where they regenerate ATP from ADP to sustain dynein-driven beating[55]. This shared domain could suggest that RIIAD1 may interact with similar ciliary partners or participate in related regulatory pathways. Notably, the reduced expression of RIIAD1 in the HS disease sample could point towards a role in maintaining ciliary energy homeostasis or motility and makes it a compelling target for further investigation. Reduced RIIAD1 expression may potentially result in less beating efficiency, which could support the congestion taking place in HS. It should be noted that only one HS disease sample was included in the present investigation, limiting the ability to draw accurate conclusions from the findings. The expression pattern was, however, stable across the three control samples, suggesting that the observed alterations are not due to interindividual variations. Further studies with more HS disease samples are needed to confirm the role of the investigated proteins in the disease.

Because previous scRNA-seq studies have established that both ciliated and non-ciliated cell counts are lower in HS compared to healthy controls[3], our findings may be indirectly related to cilia disruption and not a decrease due to a pathophysiological process affecting the expression of these proteins. FOXJ1-positive cells were also lower in HS after correcting for epithelial width. In fact, HS is a fibrotic disease, but direct evidence linking the loss of FOXJ1 expression to the HS disease state remains limited. It has however been shown that in disease respiratory tissues, cells with a fibrotic phenotype show a greater decline in ciliogenesis capacity and a marked loss of FOXJ1-positive cells[56], suggesting a link between different fibrotic diseases and cilia function. Together, these findings illustrate how systematic protein mapping in the FTE can provide valuable molecular insights into HS, a condition in which ciliary structure and function are disrupted, and may help identify proteins involved in disease-associated ciliary impairment.

On a final note, the HPA is a dynamic database that is continuously refined and expanded. The present work is based on Version 23 of the HPA resource. For example, at the publication date of this article, the HPA has advanced to version 25, with the number of FT-elevated genes increasing slightly from 310 to 313, which is a minimal change and confirms the consistency of our study across versions.

To conclude, we have integrated single-cell omics and MS data with an IHC workflow to describe the characteristic protein repertoire of FT and their motile cilia. Our IHC approach allows us to identify protein localization with a resolution sufficient for mapping down to different subcellular compartments of motile cilia. Further research on the cilia-associated proteins identified in this study could provide significant insights into the field of cilia biology, as well as the underlying mechanisms of idiopathic infertility, ciliopathy[57], and other disorders related to the FT, the reproductive system, and impaired cilia function.

## Methods

### Tissues and ethical declarations

The research complies with all relevant ethical regulations, and the study protocol has been approved by the Uppsala Ethical Review Board, ref #2002-577, 2005-338, 2007-159, and #2011-473.

Anonymized human samples were obtained from the Department of Pathology, Uppsala University Hospital, Uppsala, Sweden, as part of the sample collection governed by the Uppsala Biobank (http://www.uppsalabiobank.uu.se/en/). Informed consent was obtained from all subjects in the study. The donors received no monetary compensation. Formalin-fixed paraffin-embedded (FFPE) tissue samples were punched to 1 or 2 mm cores from following tissues: three samples of FT (ages 25, 37, and 47), two testis (ages 25 and 34), three endometrium (ages 40, 41 and 43), two cervix (ages 27 and 37), two caudate (male age 37 and female age 64), one choroid plexus (female age 19), two bronchus (males ages 75 and 82), two nasal mucosa (males ages 13 and 48), two epididymis tissues (ages 18 and 26), and assembled into a tissue microarray (TMA). See the following for more detailed information[58]. Four additional samples (three normal controls and one hydrosalpinx sample) were also utilized for the validation of candidate proteins and to explore the expression pattern in independent samples and disease. These de-identified tissue samples corresponding to the FT ampulla were obtained from the University of Michigan's Reproductive Subject Registry and Sample Biorepository (RSRSR)[59]. Only tissue sections according to the number of analyzed proteins were obtained; the remaining parts of the samples are at RSRSR. The protocols and procedures of RSRSR have been approved by the Institutional Review Board of the University of Michigan Medical School (IRBMED), registered under IDHUM00125627. Informed consent was obtained from all subjects in the study. The donors received no monetary compensation. Clinical features and descriptions of each sample are found in Supplementary Data 6.

## HPA RNA data summary

RNA expression data were retrieved from the HPA database by accessing https://v23.proteinatlas.org. The normalized consensus transcript expression levels (nTPM) were obtained by applying a normalization pipeline, as described previously[21,60]. The genes were categorized by their tissue/cell type expression level specificity as follows; at least four times higher nTPM level in a particular tissue/cell type than all other tissues/cell types (tissue-/cell type-enriched), a group of 2–5 tissues/cell types with at least four times the average nTPM levels than any other tissue/cell type (group-enriched), and a group of 1-5 tissues/cell types with at least four times the mean nTPM level of other tissues/cell types (tissue-/cell type-enhanced). These three categories together are also termed tissue-elevated. The remaining non-elevated genes with an average nTPM ≥ 1 in at least one tissue/cell type were defined as low tissue/cell type specificity, and genes with average nTPM below 1 in all tissues/cell types were considered as Not detected. The gene expression data was also categorized by the detection distribution of all genes in the tissues/cell types as follows; in a single tissue/cell type (detected in single), in more than one but less than one third of tissues/cell types (detected in some), in at least a third but not all tissues/cell types (detected in many), and in all 51 tissues/76 cell types (detected in all).

## Single-cell RNA-seq data integration

Three publicly available scRNA-seq datasets analyzing human FT tissues were downloaded from Gene Expression Omnibus with GEO accession numbers: GSE139079[37], GSE151214[5], and GSE178101[3]. (It should be noted that GSE178101 is included in the single-cell RNA resource in the HPA). These datasets were integrated and analyzed using the Seurat[61] package (Version 4.1.1). The different clusters obtained were manually annotated and identity assigned based on protein markers targeting the major FT cell types, including ciliated cells (FOXJ1, CCDC17, CCDC78), secretory cells (OVGP1, PAX8, KRT7, EPCAM), fibroblasts (DCN, COL3A1, FBN1, LUM), smooth muscle cells (ACTA2, MYH11, CNN1), endothelial cells (VWF, CD34, PECAM1). Immune cells were mast cells (TPSB2), B and T cells (CD19, MS4A1, CR2, CD3E, CD4, CD8A, FOXP3, IL17A), macrophages (CD163, CD68,

MARCO, MRC1, MSR1), dendritic cells (GZMB, IL3RA), and NK cells (KIR2DL4, NCR1). One cluster was defined as unidentified, as no distinct cell identity could be assigned and, thus, was not used in further analyses. For ease of analysis, the ciliated cell clusters were merged into one main ciliated cell cluster and the secretory cell clusters into one main secretory cell cluster. All other cell clusters (fibroblasts, smooth muscle cells, endothelial cells, mast cells, B and T cells, macrophages, dendritic cells, NK cells, as well as non-defined clusters) were merged into one composite Other cluster.

## Immunohistochemical staining

Tissues were processed as previously described[62]. In summary, TMA paraffin blocks were cut in 4μm sections with waterfall microtomes (Microm HM 355S, Thermo Fisher Scientific, Fremont, CA), placed on SuperFrost Plus slides, and dried at room temperature overnight. Slides were baked at 50 °C for 12 – 24 h before immunohistochemical (IHC) staining. Slides were stained with the Autostainer 480 Instrument (Lab Vision, Fremont, CA). Primary antibodies were diluted in Antibody Diluent solution (Thermo Fisher Scientific, Freemont, CA), and antibody dilution was optimized based on the International Working Group for Antibody Validation (IWGAV) framework suitable for IHC[63], taking into consideration the correlation of antibody labeling in tissues with an antibody-independent method (bulk RNA expression), and also the specificity of an antibody compared to (an)other antibody/antibodies against the same target protein. The expected staining pattern based on the available literature review was also considered as well. All representative micrographs and IHC images were obtained from a single IHC staining per protein target on the study TMA. Antibody optimization was performed using a test TMA slide containing 20 different normal tissue types, including FT. For each antibody, optimization and titration were performed across two to three independent staining runs to identify the dilution that yielded a clear and distinct immunohistochemical signal in the organ(s) with the expected highest expression, while minimizing background staining in other tissues. This multi-stage optimization procedure enabled rigorous assessment of staining robustness and reproducibility across experimental runs. Only antibodies demonstrating consistent, specific, and reproducible staining patterns on the test TMA were subsequently applied to the study TMA. TMA slides were scanned by using the Scanscope AT2 system (Aperio, Vista, CA), equipped with a 40x objective. A detailed protocol of the automated IHC staining can be found at https://www.proteinatlas.org/about/help. All antibodies used in this study are available from Atlas Antibodies unless otherwise stated. The antibody target proteins, reference codes, Research Resource Identifiers (RRID), and dilutions are listed in Supplementary Data 7.

## Annotation

The IHC images were reviewed using ImageScope (Leica Biosystems) or QuPath[64], without any digital adjustments to brightness, contrast, or color balance. Stainings were manually annotated by visual estimation, assessing staining intensity at four levels (negative, weak, moderate, or strong) and scoring the subcellular compartments of staining (cilia, general cytoplasmic/membranous, or nucleus). The annotation in FTE cells also specified whether the staining was observed in ciliated or non-ciliated (secretory) cells, with this distinction made strictly based on morphological features within the same tissue section. For ciliated FTE cells, an extended and more detailed annotation captured whether the staining was associated mainly with the tip or upper segments of the cilia (cilia tip), the middle (cilia), or the base of the cilia (rootlet). All annotations and scoring were performed independently by two observers, who discussed and resolved any discrepancies by consensus. In Supplementary Fig. 9, a positive control (FT tissue) and a negative control staining with low bulk RNA-seq levels are presented for each antibody.

## Re-analysis of Deep Visual Proteomics data from FOXJ1-positive and FOXJ1-negative cells

Quantitative proteomic profiling data of FTE analyzed with Deep Visual Proteomics (DVP) were used in this study. DVP is an imaging-guided, AI-assisted MS workflow that enables ultra-sensitive protein quantification in FFPE tissue blocks[38]. FOXJ1, a transcription factor essential for motile ciliogenesis, was used to distinguish ciliated from non-ciliated FTE cells. Antibody-labeled FOXJ1-positive and FOXJ1-negative cells were isolated by laser microdissection, and their proteomes were analyzed by highly sensitive mass spectrometry. Differential abundance testing across ~5000 protein groups was conducted using Student's t-tests, yielding test statistics, abundance differences, and corresponding p- and q-values. From this dataset, we curated candidate proteins relevant to our study, excluding ambiguous rows with multiple protein identifications. Enrichment was defined by the t-statistic (positive values indicating FOXJ1-positive enrichment, negative values indicating FOXJ1-negative enrichment). Both significant and non-significant proteins were retained, with statistical support explicitly indicated. Proteins were annotated as present in FOXJ1$^+$, FOXJ1$^-$, or No data, and these classifications were visualized in a heatmap. The full proteomic dataset, statistical results, and annotations are provided in Supplementary Data 8.

## Data analysis, visualization, and statistics

All data analysis was done with the statistical language R (RRID:SCR_001905 versions 4.1.1 or 4.2.1). Upset plots were generated with the complexUpset R package[65]. The dot plot single-cell RNA-seq overview was generated with Seurat R package[66]. The heatmap was made with the R package ComplexHeatmap[67] and default hierarchical clustering with K-means = 5 was performed on the scRNA-seq integrated dataset. Alluvial plots and pie charts were done with the ggplot2 package[68]. Protein evidence from the UniProt resource was downloaded from UniProt on May 6th, 2022. Protein evidence and Transcript evidence were used to group the genes. The No evidence group includes the UniProt entries Inferred from homology, Predicted, and Uncertain. For one gene, AC117457.1 (ENSG00000275163), there was no UniProt data due to a readthrough transcript. The Gene Ontology treemap was done by creating a synthetic hierarchy by aggregating the GO terms based on word similarity. The GO terms with the highest overlap of genes and lowest p-value were set as the so-called parent term and are visualized on top of all underlying GO terms. The analysis was done by the R package rrvgo[69] to calculate the semantic similarity between words and subsequent aggregation. The R package treemapify was used to generate the plot. The lists of ciliopathy-related genes were downloaded from CiliaMiner (https://kaplanlab.shinyapps.io/ciliaminer/) on December 03, 2022[34]. The list SysCilia Gold Standard V1[28] was downloaded on February 02, 2022, from http://www.syscilia.org/goldstandard.shtml. The list SysCilia Gold Standard V2 was downloaded from the supplementary PDF file in the related publication[27] and converted manually into a table format for subsequent analysis in R. All other cilia-related gene/protein lists were retrieved from the supplementary files of the referenced publications. Comparisons to cilia-, flagella-, and ciliopathy-related protein lists were plotted with the R package plotrix[70].

Image analysis of FOXJ1 IHC staining was performed in QuPath (version 0.6) by segmenting cells using the InstanSeg tool and default settings. A single-measurement object classifier was performed to generate a threshold value for positive and negative cells, i.e., the presence of DAB (FOXJ1-positive cells) or the absence of DAB (FOXJ1-negative cells). To quantitatively assess the width of the epithelium in the additional control samples and hydrosalpinx sample, annotated epithelial tissue regions were exported from QuPath. Epithelial regions were manually annotated in QuPath, with annotations drawn closely following the epithelial mucosa and tracing the basement membrane. To exclude luminal voids, a simple pixel classifier was applied within QuPath to distinguish white (void) areas from tissue, ensuring that only epithelial mucosa was retained for subsequent analysis. Tissue regions and cells with FOXJ1 status were exported from QuPath in GeoJSON format as vector polygons, and hereafter, all analyses were performed in Python using geopandas, shapely, scikit-image, scipy, networkx, matplotlib, and in-house scripts (available upon request). In cases where multiple tissue polygons were present, the largest connected region was selected for analysis. Each polygon was rasterized, and the epithelium area from QuPath was converted to a binary mask, then used to compute a Euclidean distance transform (EDT). This encodes the distance of every interior pixel to the nearest boundary (lumen or basement membrane). The epithelium mask was also subjected to skeletonization (morphological thinning) to produce a one-pixel-wide medial axis that represents the geometric centerline of the tissue ribbon. To ensure that the centerline traversed the ribbon in a single pass from one end to the other, the skeleton was converted into a graph and analyzed using graph-theoretical algorithms. The longest path between terminal nodes (endpoints) was extracted, representing the main axis of the epithelium. At each point along this centerline, the local width of the epithelium was estimated directly from the EDT values, defined as twice the distance from the centerline point to the nearest boundary. These measurements were mapped back to the original polygon coordinates, preserving the geometric context, and epithelial width values were expressed in both pixels and micrometers, using a calibration factor of $0.5\,\mu m$ per pixel. The epithelium outline was overlaid with the centerline, pseudo-colored according to local width values. In addition, a width profile was generated by plotting the local width as a function of cumulative arc length along the epithelium. Per-point measurements were exported in tabular format, and descriptive statistics (mean, median, first and third quartiles, min and max) were calculated to summarize epithelial width distributions.

## Statistics and reproducibility

No statistical method was used to predetermine sample size. No data were excluded from the analyses. The experiments were not randomized. The Investigators were not blinded to allocation during experiments and outcome assessment.

## Reporting summary

Further information on research design is available in the Nature Portfolio Reporting Summary linked to this article.

# Data availability

The immunohistochemical imaging data generated in this study are available in the Human Protein Atlas (HPA) at v25.proteinatlas.org, searchable by gene name in the Tissue Resource. These images are also available via BioStudies under accession code S-BIAD1031. The bulk RNA expression data used in this study are available in the HPA database at v23.proteinatlas.org and can be downloaded directly. The single-cell RNA-seq data used in this study are available in the GEO database under accession codes GSE139079 [https://www.ncbi.nlm.nih.gov/geo/query/acc.cgi?acc=GSE139079], GSE151214 [https://www.ncbi.nlm.nih.gov/geo/query/acc.cgi?acc=GSE151214], and GSE178101 [https://www.ncbi.nlm.nih.gov/geo/query/acc.cgi?acc=GSE178101]. The mass spectrometry proteomics data used in this study are available in the PRIDE database under accession code PXD023904 [https://www.ebi.ac.uk/pride/archive/projects/PXD023904]; the subset used in the present study is provided in Supplementary Data 8. Source data are provided with this paper.

# Code availability

All source data and analysis scripts required to reproduce the figures are available at https://github.com/LindskogLab/Spatial-map-of-cilia-associated-proteins-in-the-human-fallopian-tube.

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

## Acknowledgements

This study was supported by the Knut and Alice Wallenberg Foundation (#2015.0344). CL was also supported by the Swedish Research Council (2022-02742). We want to direct our sincere thanks to Jonas Gustavsson, Borbala Katona, and Rutger Schutten, as well as other staff members of the Human Protein Atlas, for producing the TMA and immunohistochemical staining and imaging. JNH was supported by a Postdoctoral Fellowship from the Wenner-Gren Foundations, Sweden, and by an EMBO Postdoctoral Fellowship (ALTF 556-2022).

## Author contributions

Conceptualization: C.L., F.H. and L.M.; Data curation: F.H. and A.D.; Formal Analysis: F.H., A.D., L.M. and J.N.H.; Funding acquisition: C.L. and M.U.; Investigation: F.H., A.D., L.M., J.N.H. and C.L.; Methodology: C.L., F.H. and L.M.; Project administration: F.H., A.D. and C.L.; Resources: L.M., J.N.H. and S.B.S.; Software: L.M., F.H., and J.N.H.; Supervision: C.L., L.M., E.L., M.U. and M.O.; Validation: F.H., L.M. and C.L.; Visualization: L.M., F.H., A.D., R.S., J.N.H.; Writing – original draft: F.H. and C.L.; Writing – review & editing: A.D., J.N.H., M.O. and L.M. All authors have read and approved the final version of the manuscript.

## Funding

## Competing interests

The authors declare no competing interests.
