## [Transparent Peer Review file · Nature Communications]

A high-resolution spatial map of cilia-associated proteins in the human fallopian tube

Corresponding Author: Dr Cecilia Lindskog

Version 0:

Reviewer comments:

Reviewer #1

(Remarks to the Author)

In this study, Hikmet et al. aimed to identify cilia-associated proteins in human fallopian tubes compared to other tissues with ciliated epithelial cells such as the respiratory tract, endometrium, choroid plexus, epididymis, testis (sperm flagella), etc. For validation, the authors also harbored publicly available scRNA-seq data from the literature, mass spectrometry data from laser micro-dissected FOXJ1-positive and -negative cells in human fallopian tubes, and proteins annotated in multiple datasets. To this end, the authors were able to identify novel proteins that are present in the cilia with no prior transcript or protein data. In addition, some of the proteins were spatially located at the subcellular levels. Some of these proteins were specifically “enriched” in the ciliated epithelial cells of the fallopian tube. However, they are not “exclusively” expressed in the fallopian tubes. There are multiple strengths in this study, including the use of three validation methods and the cross-listing of the identified proteins with the ciliopathy database. More importantly, the IHC analysis datasets from this study will be publicly available online. The conclusions were supported by the results. The methods are well described. In addition, the limitations of this study were discussed. Although this work used novel approaches, the overall findings are primarily descriptive and focus on characterizing protein expression. There are significant weaknesses listed below:

1. As mentioned above, this study is descriptive and lacks functional testing of some of the ‘unique’ proteins identified from this study.
2. Since the fallopian tube is the reproductive tissue, it is unclear why samples from patients aged 74 and 85 are included – the authors need to incorporate the rationale and appropriate justification for this inclusion.
3. Most statements made in the introduction require citations. This needs to be addressed.
4. The results section is too long and includes information that could be described in the discussion. These are some of the texts that explain Figures 2 and 3. The interpretation should be included in the discussion only.
5. Negative controls need to be included in the image in all IHC staining.
6. It was mentioned in the introduction that the identified proteins within the reproductive system could serve as avenues for the design of novel treatment options or drug targets. However, how the findings from this study can be implemented is unclear since the proteins are enriched in ciliated cells of the fallopian tubes. Still, proteins are also “expressed” in other tissues. – this needs to be included in the discussion.
7. Information in Fig. S3B needs to be included in the main results section as the manuscript highlights its clinical translationability. The information on the overlap of proteins identified with the ciliopathies database should be included in the main figure.
8. Fig. S3B, CFAP45 showed a matching with three different types of ciliopathies. However, the manuscript did not mention or discuss the importance of this protein in reproductive function, i.e., do patients with the mutation in the CFAP45 gene have reproductive defects? This could also be functionally tested using a conditional knockout mouse model, specifically in the fallopian tubes.

Reviewer #2

(Remarks to the Author)

This report takes a deep dive into the proteome of ciliated cells of the fallopian tube. Proper functioning of ciliated cells in tubal epithelia is essential for reproduction, and so a more comprehensive understanding of key structural and functional proteins in these cells is important for fertility research.

The manuscript is rooted in re-analyses of data from the Human Protein Atlas. The authors create a low-density TMA and stain for 130 proteins nominated, to understand whether the cilia-associated proteins are unique to the fallopian tube ciliated epithelium or are expressed by other ciliated cell types. The authors also mine published single cell expression data sets, a spatial single-cell proteomic dataset, and The CiliaCarta compendium to curate a comprehensive set of fallopian tube cilia associated proteins and annotate the subcellular localization of those proteins.

This report is clearly presented, easy to follow and well written. It is an impressive descriptive study but would benefit substantially from functional validation – for example, if any of these proteins are depleted in in vitro or in vivo models, does that impact cilia function or survival of ciliated epithelial cells? Insight into which (if any) of these proteins exhibit dysregulated expression in tubal pathologies would help prioritize which proteins are most important to study for human health and reproduction.

Minor point

- Figure S2, the authors should consider including expression of cell-type defining markers to support cell type assignments

Reviewer #3

(Remarks to the Author)

This research aims to identify new proteins associated with cilia in the human Fallopian Tube (FT) by using a comprehensive approach that combines high-throughput transcriptomics (both bulk and in silico single-cell RNA sequencing) and proteomics (using immunohistochemistry).

The authors' recognition of the significance of expanding knowledge and investigating less explored areas is noteworthy. This study examines molecular differences (at the RNA and protein levels) in a vital organ for women's reproductive health—the fallopian tube—.

However, a notable concern arises regarding the utilization of in silico data and public datasets for selecting candidate genes to validate within the chosen sample set for the study. Specifically, relying on three single-cell RNA sequencing datasets to evaluate the cell specificity of the identified genes through bulk analyses may introduce bias and may not lead to robust conclusions. Consequently, the resulting study primarily provides descriptions and lacks functional reproducibility, as the data come from external sources rather than being internally generated.

Introduction

1. While authors perform an extensive description of the motile cilia, it is only supported by few general references. Please, add literature supporting specific mechanisms such as the active movement of a motile cilium (page 3, lines 58-59), dynein arms (lines page 3 60-62).

2. Page 3, lines 68-69: The statement “The controlled import and export to and from cilia generates a unique protein repertoire in cilia” may suggest that such repertoire is well studied and established as a standard reference. However, it is crucial to provide clear interpretation and proper referencing on the extent of existing studies in this area, especially since the primary aim of the paper is to pinpoint and elucidate essential proteins associated with ciliary function.

3. Page 4, lines 87-88: please, ensure to include in this section the most relevant “other tissues with cells employing motile cilia” that were analysed in this study and contrasted with FT.

Results

Overall, I find the results section well-written and comprehensive. However, I noticed a missing section detailing the experimental workflow employed in this study. It would greatly enhance the clarity and understanding of the paper if you could provide a concise description of the experimental workflow, outlining the sequence of steps and methodologies utilized in your research. Additionally, I would advise you to consider preparing a figure illustrating the experimental workflow. Such a visual aid would significantly improve the readability and comprehension of your manuscript for the reviewers and the readers.

The FT-elevated transcriptome

1. Page 6, Lines 103-108: the elucidation regarding the HPA project classification system used in this study should be included in the Methods section under “HPA RNA data summary”. Combining this information with the details provided in the Methods section concerning the gene categorization based on tissue type expression specificity (page 20, lines 479-485) would be beneficial. Consolidating these explanations within the same section would enhance clarity and coherence.

2. If the creation of the FT-specific gene set involves its inclusion in the bulk analysis, what is the reasoning behind the decision behind excluding the ciliated cells from the FT in the HPA scRNA-seq dataset?

3. Page 7, lines 136-37: Could you please provide the data for which two genes had uncertain evidence?

The spatial localization of FT-elevated proteins in ciliated and non-ciliated cells

1. Page 7, lines 149-150: please provide more explanation on what the reliability scores “Approved, Supported and Enhanced” mean.

2. Page 8, line 183: please rephrase the sentence to clarify whether the total count of identified proteins is 29, with 7 being unique to the cilium base, or if out of 36 proteins, 29 are common and 7 are specific to the cilium base.
3. Page 9, lines 191-197: Although illustrative for the proteins mentioned previously, this paragraph is only referring to model organisms. As mentioned by the authors there is a knowledge gap regarding the level of evidence we have for some proteins, but consider rephrase this paragraph, considering that transcripts and proteins are not always conserved between animal models and human.

Spatial analysis of cilia-associated proteins in other tissues with motile cilia.

1. Page 11, line 240-249: in this paragraph the authors discuss the proteins detected across various analysed tissues. They provide insights into the known functions of some proteins (e.g. GAS2L2 or CRTAC1) while indicating if others are uncharacterized (e.g. KIAA2012). However, no information is provided for CFAP73, PNOG and SLC27A6. Please, confirm if these proteins are characterized to maintain consistency with the provided list.

Validation of FTE cell specificity

1. Page 12, line 283: the Deep visual Proteomics (DVP) workflow is first mentioned here, however, no citations or explanations regarding this approach are provided in the Methods section. It is recommended to include comprehensive information about this algorithm in the Methods section to ensure clarity and understanding.
2. Page 13, lines 301-307: Consider moving this paragraph to the discussion, since it does not provide any results per se.
3. Page 13, lines 308-315: Considering the various tools and resources employed during this validation step, is there any evidence from in vivo studies or protein-protein interactions indicating the involvement of the examined proteins in ciliary function? The credibility of results, their interpretation, and extrapolation heavily rely on the level of evidence in such studies. I strongly suggest conducting a re-analysis where the level of evidence for each gene/protein is treated as a variable.
4. Page 13-14, lines 317, 329, 339: in these lines there are data presented as fractions of the total of proteins characterized: "a quarter of the proteins", "two thirds of the 130 proteins", "a sixth", "an additional eight". This is a strange way of presenting data; I suggest providing the precise numerical values instead.

Discussion

1. Page 15, lines 348-350: Concerning the subcellular localization of the proteins, emerging techniques such as spatial transcriptomics or proteomics are gaining traction, offering potential advancements in resolution and reproducibility at the single-cell or single-spot level. Please, provide the rationale for having chosen IHC over these novel techniques, especially considering their potential for enhanced resolution, reproducibility, and quantitative analysis. This rationale becomes even more crucial when studying the ciliated epithelium, a histological subtype that demands the utmost precision to achieve results at the subcellular level.
2. Page 10, lines 217-218: In accordance with the authors' findings, samples of the testis have been incorporated into the study to facilitate a comparison of expression and localization between cilia and flagellum. However, I find the discussion on this comparison somewhat lacking, particularly regarding the genes expressed in both types of structures, especially considering whether this tissue was included specifically for this purpose.
3. Page 16, line 382, page 16: Please, express data as an exact number, not as a fraction ("a fourth of the 130 proteins").
4. Page 18, lines 424-429: I suggest providing additional information about intercalary cells since the authors highlight their significance within the FTE. It would be interesting to include further data describing the protein expression on these cells even if it is not specific to them. Additionally, it would be valuable for the authors to propose potential reasons for the absence of proteins specifically localized to intercalary cells.
5. Regarding the limitations of the study, it is noteworthy to mention that the sample size for fallopian tube tissue samples is relatively small, comprising only five samples, which may limit the generalizability and robustness of the findings. Apart from that, there is only three samples from endometrium, one sample from choroid plexus, and two samples for the rest of the tissues remaining (testis, caudate, bronchus, nasal mucosa and epididymis), that might not be sufficient to raise strong conclusions about the gene and protein expression in the studied subjects. Please, consider adding more samples to the sample size.

Methods

Tissues and ethical declaration

This section should address a more elaborated description and codification of both the donors and the samples.

1. Additional information regarding the cohort description (age, potential infections, surgical indication, menstrual cycle in the case of reproductive tissues) is required, as they could impact the features and function of the analysed tissues. It is also crucial to include details regarding the specific criteria used for inclusion and exclusion, as well as the methods utilized for determining these criteria. Please, provide a supplementary table clarifying these details while maintaining donor anonymity. Regarding the acquisition of tissue specimens, please, provide details on the sample collection process, the age of the blocks, and any potential discrepancies in processing fresh tissue into formalin-fixed paraffin-embedded (FFPE).
2. Page 20, lines 470-475: What is the significance of the "y" character accompanying the number for each sample type? This is somewhat perplexing as it could be interpreted as either indicating the age of the FFPE block, the age of the tissue donor, or serving as an anonymized code for the sample. If it is meant to denote the anonymized code, it appears confusing that donor 25y has contributed both fallopian tube tissue and testis tissue. Please provide clarification on this matter and assign appropriate anonymized names to the samples.

HPA RNA data summary

1. Page 20, line 478: As noted earlier, incorporating the explanation of the HPA project classification system outlined in the Results section (page 6: lines 103-108) could enhance comprehension and serve as an introductory guide to the categories detailed subsequently.
2. It would be also helpful including a summary table that collects the categories discussed in the text along with their corresponding explanations. This will offer readers a concise representation of the different classifications thereby enhancing clarity and understanding of the content.

Single- cell RNA-seq data integration

1. Despite the integration of the three publicly available single-cell datasets is acknowledged, there is a lack of information regarding the specific integration among bulk RNAseq and single-cell RNAseq data. Please, add this information for clarity.
2. Page 21, lines 489-500: please, provide specific references to single-cell atlases or available datasets that were used when referring to "well-known protein markers of the fallopian tube cell type".

Annotation

1. Page 22, lines 529-530: how were the four levels of staining intensity determined? Was there a quantitative method employed for these determinations? Please, specify the image processing software used for managing the IHC results, along with the relevant parameters for image evaluation or the statistical analysis conducted.
2. Page 22, line 532: how was the distinction between "ciliated and non-ciliated cells" determined? Please indicate whether there was assessment conducted by a certified pathologist or if there was a double observer approach to confirm these findings.

Data availability

- Page 24, line 574: It seems that the provided link www.v21.proteinatlas.org is not working correctly. Please, provide a valid link.

Figures and legends:

Typically, figures should be comprehensible solely by referring to their accompanying legends. However, the current figure legends are insufficiently detailed to be self-explanatory. It is recommended to either enhance the clarity of the figures and to improve the comprehensiveness of the figure legends.

- Figure 1B: This representation of a GO enrichment seems a bit confusing, and the smaller boxes do not have enough resolution to for effective reading. Exploring other visualization options for GO enrichment plots that incorporate statistical values such as p-values, z-scores, or defined thresholds would be beneficial in ensuring the enriched functions are more easily discernible and comprehensible.
- Figure 1C: Similar to Figure 1B, I recommend considering alternative plots such as bar charts, stacked bar charts, or line graphs, which provide clear numeric references for each category and facilitate easier comparison between categories. Implementing such plots is likely to enhance the clarity and understandability of your data visualization.
- Figure Legend 1: The abbreviations within section (B) of the legend should be relocated to section (C) of the figure, as they pertain to that part of the illustration.
- Table S1: The category "Detected in fallopian only" in the table is labelled as "Detected in single" in the legend. Please ensure uniformity by using the same terminology in both the legend and the table.
- Figure 2A: This figure seems somewhat unclear; a more detailed explanation of the elements depicted in the graphics above and to the right, as well as clarification on the numerical values associated with them, would greatly enhance its clarity.
- Figure 2B: Please specify in the figure legend the type of tissue shown.
- Figure 3A: Within the intersection plots, there are two types of bars: (1) vertical bars with numerical values above them and (2) horizontal bars representing subcellular compartments, both accompanied by numerical annotations. Please indicate in the legend what specifically "numbers" indicate from the figure.

Minor comments:

1. Page 8, line 159: FTE, please specify acronyms when first mentioned.
2. Page 8, line 167: please, replace the expression "well studied" by a most concise concept and consider incorporating levels of evidence (level of evidence in vivo / in silico, curated/ non-curated, etc...) where applicable.
3. Page 10, line 211-212: please provide the proper reference for this sentence.
4. Page 11, line 248: please specify of which tissue is OVG1 a well-known marker.
5. Page 12, line 278: please, separate "whereof" to "where of"
6. Page 14, lines 335-337: please provide the proper reference for these sentences
7. Page 20, line 472: Given the coding system applied to all patients, it's possible that "male 37" might be a typo error and actually refers to "male 37y", correct?
8. Page 21, line 514: IWGAV, please specify acronyms when first mentioned.

Reviewer #4

(Remarks to the Author)

Review of "A high-resolution spatial map of cilia-associated proteins based on characterization of the human fallopian tube-specific proteome" by Hikmet et al.

Background and importance: Ciliated cells in the human body play a very important role in human physiology and disease. As such, the main aim of the authors to add to the collection of specific genes expressed in ciliated cells is important and laudable. However, the execution of their plan leaves much to be desired, to the extent that I cannot recommend publication, at least not in anything like the current forms.

What the authors did: This paper mines data from the Human Proteome Atlas, a very useful community resource that the authors are leading or part of. Despite the name, it contains much RNA expression data, from which the authors start to identify genes with elevated expression in the fallopian tube compared to other tissues. They found 315 genes categorized as "tissue-enriched", "group-enriched", or "tissue-enhanced" in the fallopian tube. Gene Ontology analysis confirms that a majority of these genes are associated with motile cilia function and sperm/flagellar motility. Out of the 315 genes, the authors then selected 130 genes with reliable antibodies available from the Human Protein Atlas for further immunohistochemistry (IHC) profiling. If I understand correctly (not very clear in the manuscript), the sole experimental work of the paper is then the in-depth IHC analysis on a tissue microarray containing fallopian tube and other tissues with motile ciliated cells (e.g., respiratory tract, brain ventricles). The purpose is to map the subcellular localization of the 130 proteins, identifying their association with ciliated cells, non-ciliated cells, or specific ciliary compartments. The authors then consult three sources of single cell information to validate their IHC findings using: a. Single-cell RNA-seq data from fallopian tube epithelial cells b. Mass spectrometry data from laser-captured ciliated (FOXJ1+) and non-ciliated (FOXJ1-) fallopian tube cells and c. Comparison with existing cilia databases (CiliaCarta, SysCilia, etc.).

An obvious critique of this effort is that it is entirely descriptive: No functional experiments are performed and no new biological insights are forthcoming. That by being the case, the authors would have had to have a very solid resource outcome, i.e. one would have expected a golden set of proteins that the cilia community can now study. Unfortunately, this is not the case.

The initial start from bulk RNAseq is questionable as this involves at least two major epithelial cell types that are mixed (plus some others), making it difficult to do the enrichment analysis. The authors should come up with a more inclusive list of proteins from more of their sources (and the recent single cell atlases?). They should then devise a logical scheme of how to reduce this catalogue to a smaller set of high confidence proteins that fulfill some combination of evidence. Maybe a score with some rational basis instead of a purely subjective discussion, ideally even some rudimentary machine learning based on the multiple data sets. In any case some elementary statistics is necessary. Some functional validation would of course add tremendously to the paper although the reviewer realizes that this is easier said than done.

The novel proteins should be discussed as for their role in ciliated cells. It should be made clear what was known before and what is novel after this paper. In general, while the authors provide some biological interpretation, they could further elaborate on the potential functional roles and mechanisms of the novel cilia-related proteins they identified, based on literature and bioinformatics analyses. They should explore the potential implications of their findings for understanding ciliopathies, infertility, and other disorders related to cilia dysfunction, and discuss how their data could inform future studies in these areas. The authors could discuss the limitations of their study, such as the potential for missing low-abundance proteins or proteins with poor antibody performance, and how these limitations could be addressed in future work.

Specific comments

1. The IHC images shown in this manuscript are a valuable resource for researchers interested in the spatial localization of FT-related proteins. Due to figure limitations, not all staining images can be shown in the main publication. While the IHC images will be available at the Biostudies repository, an easy-accessible database or extended supplementary file may be helpful for the community.

2. In Figure 2B and the corresponding text, a localization of proteins mainly to the nucleus of ciliated FTE cells is described. Visually, it is correct that the proteins are located in the nucleus by staining, but they also seem to be visible in the cytoplasm and cilia (e.g. CAPS). In contrast, FOXJ1 is exclusively located in the nucleus. EFCAB1 seems to be also positive in the nucleus, but is associated to the group of cytoplasmatic markers. The distribution should be sub-grouped (e.g. exclusive nuclear, nuclear + cytoplasm etc.). Also sentences such as 'localization to mainly ciliated FTE cell nuclei' could be adapted accordingly.

3. Line 245: After discussing the results of Figure 3B, the authors come back to results displayed in Figure 3A. A reference to Fig. 3A is missing here and the authors may consider re-arranging this section.

4. Line 251: The IHC result of SLC27A6 is described and referred to Figure 3B. However, the staining image is not displayed here.

5. Ciliated cells from FT were not included in the HPA scRNA-seq dataset, but the authors re-visit datasets from Hu et al., Dinh et al. and Ulrich et al. There is another single-cell resolved atlas on the post-menopausal fallopian tube by Lengyel et al. (PMID: 36543131) published in 2022. It would be interesting to see this dataset integrated as well and to elaborate on the pre- and post-menopausal state of the fallopian tube for the most relevant proteins of your dataset. This could be of relevance as Hu et al. describes post-menopausal FT, while Dinh et al. shows all (but one) premenopausal FT.

6. Figure 4A: Based on the integration of the three scRNA-seq datasets and mapping the resulting clustering on your FT-specific proteins, the authors may discuss in more detail the difference between the ciliated 1 vs ciliated 2 population. Similarity to secretory cells is indicated, but is there a biological translation of the findings? If the expression signal was

consistently higher in the population of ciliated 1 cells, could this be a systematic technical effect?

7. In Figure S3B, the authors depict ciliopathy-related genes and conclude that a sixth of the 130 genes is associated to misfunctions of the cilia. It would be interesting to discuss examples and how this dataset could contribute to suggest targets to treat or understand ciliopathies.

8. Infertility is mentioned as a main motivator to study the fallopian tube. The readership would profit from a more detailed analysis and discussion on fertility-associated genes and how this study is contributing to understand the pathways disentangling pathways involved in infertility as written in the abstract.

9. Apart from the above comments, the figures are very nicely and professionally done.

Minor comments

1. Figure 1C: evidence and GO term level are difficult to separate in figure. A visual clarification could help.

2. Line 155 to line 158: "proteins made up the largest proportion of the 130 selected proteins. Is this referring to protein subunits and isoforms or rather quantitative level/abundance? Please explain the term "proportion" in more detail.

3. Figure 2B, Figure 3A: (representative) scale bars of microscopy images are missing.

4. Figure 4B: the color of the grey grid lines interferes with the readability of database labels. Opaque grid lines could improve the readability of the figure content.

5. Line 378: 'non-FOXJ1-positive' - rephrase to FOXJ1 negative

Version 1:

Reviewer comments:

Reviewer #1

(Remarks to the Author)

The authors have addressed my previous concerns and included additional experiments involving a hydrosalpinx sample. They have also rewritten certain sections of the manuscript. The data and figures are of outstanding quality. Although the findings are still primarily descriptive and lack functional analysis, they offer valuable insights into the field of cilia biology. The added data has significantly enhanced the overall quality of the manuscript. There are only minor points to be addressed regarding the hydrosalpinx samples.

1) Please clarify whether the regions obtained for figures 7C-D and S8 were derived from the fimbria for all samples, including controls and hydrosalpinx.

2) Please acknowledge in the discussion that the hydrosalpinx sample was derived from an n of 1; therefore, it is challenging to draw accurate conclusions from this finding. This reviewer understands the difficulties in obtaining hydrosalpinx samples. However, this caveat must be presented to the readers.

Reviewer #2

(Remarks to the Author)

This is a valuable resource of marker expression for the community of researchers working on the fallopian tube and pathologies impacting the ciliated cells of this organ, and goes well beyond the scope of prior studies in its depth. In addition the validation of a large panel of antibodies will serve as an additional research resource for researchers. The team did a solid job of responding to reviewer comments, although the inclusion of just a single specimen impacted by hydrosalpinx does limit the conclusions that can be made from that analysis.

Reviewer #3

(Remarks to the Author)

The authors have addressed satisfactorily my suggestions

Reviewer #4

(Remarks to the Author)

The authors have done a commendable job and substantially improved the manuscript in response to the reviewers' concerns. I appreciate the considerable effort that went into this revision. That said, the only revision document made available to this reviewer was a gigantic, merged pdf, and no track changes document.

Regarding functional validation (brought up by all reviewers): The addition of the hydrosalpinx (HS) analysis is a meaningful step beyond pure description. By examining their top 11 uncharacterized proteins in a disease context and identifying three candidates (FHAD1, RIIAD1, C2orf81) with reduced expression in HS tissue, the authors have demonstrated potential clinical relevance. The quantitative analysis of FOXJ1+ cell density and epithelial width measurements adds rigor to these observations. While this does not constitute mechanistic validation, it provides the community with testable hypotheses.

Regarding data organization: The reorganization of the 123 ciliated-cell proteins into seven evidence-based groups (Figure 3) is a helpful addition that allows readers to quickly identify which proteins are truly novel versus those with existing characterization. This partially addresses my concern about providing a "golden set" of proteins for the community to study.

Remaining considerations: My original concern about the bulk RNA-seq starting point mixing cell types was not directly addressed, though the extensive validation against scRNA-seq and mass spectrometry datasets mitigates this issue in practice. I had also suggested a more rigorous, potentially ML-based scoring approach rather than subjective grouping; the authors' seven-tier system is reasonable but remains somewhat arbitrary. The sample size for the HS analysis (n=1) limits the strength of conclusions, though I acknowledge this as an exploratory addition.

Despite these quibbles, is now a well-validated spatial protein resource that fills a genuine gap in our understanding of fallopian tube and motile cilia biology. The IHC dataset covering 133 proteins with subcellular resolution, cross-validated against multiple orthogonal methods, will be valuable for researchers studying ciliopathies and reproductive disorders. The data will be publicly available through the Human Protein Atlas, ensuring community access.

Reviewer #1 (Remarks to the Author):

In this study, Hikmet et al. aimed to identify cilia-associated proteins in human fallopian tubes compared to other tissues with ciliated epithelial cells such as the respiratory tract, endometrium, choroid plexus, epididymis, testis (sperm flagella), etc. For validation, the others also harbored publicly available scRNA-seq data from the literature, mass spectrometry data from laser micro-dissected FOXJ1-positive and -negative cells in human fallopian tubes, and proteins annotated in multiple datasets. To this end, the authors were able to identify novel proteins that are present in the cilia with no prior transcript or protein data. In addition, some of the proteins were spatially located at the subcellular levels. Some of these proteins were specifically “enriched” in the ciliated epithelial cells of the fallopian tube. However, they are not “exclusively” expressed in the fallopian tubes. There are multiple strengths in this study, including the use of three validation methods and the cross-listing of the identified proteins with the ciliopathy database. More importantly, the IHC analysis datasets from this study will be publicly available online. The conclusions were supported by the results. The methods are well described. In addition, the limitations of this study were discussed. Although this work used novel approaches, the overall findings are primarily descriptive and focus on characterizing protein expression. There are significant weaknesses listed below:

We thank the reviewer for the positive feedback and constructive comments regarding our manuscript. The specific comments have been addressed below.

1. As mentioned above, this study is descriptive and lacks functional testing of some of the ‘unique’ proteins identified from this study.

This is an important point. To add functional validation to the descriptive data, we have now performed IHC staining of our top eleven uncharacterized cilia proteins in a patient sample with a hydrosalpinx diagnosis (details in Table S7).

2. Since the fallopian tube is the reproductive tissue, it is unclear why samples from patients aged 74 and 85 are included – the authors need to incorporate the rationale and appropriate justification for this inclusion.

We agree with the reviewer that it is not clear why samples from patients aged 74 and 85 were included. These samples have been removed, and all scoring and representative images have been revised based on the included samples.

3. Most statements made in the introduction require citations. This needs to be addressed.

We have added relevant citations to the various statements of the introduction in the revised version of the manuscript.

4. The results section is too long and includes information that could be described in the discussion. These are some of the texts that explain Figures 2 and 3. The interpretation should be included in the discussion only.

We agree with the reviewer that the results section is very long. In the revised version of the manuscript, we have moved parts with interpretative text in the results section to the discussion.

5. Negative controls need to be included in the image in all IHC staining.

Negative controls for all antibodies used in IHC stainings have been assembled in Figure S9 in the revised submission. The negative controls are shown in the form of IHC stainings in tissues that, based on mRNA expression levels, are expected to lack or have less abundant expression of the target protein.

6. It was mentioned in the introduction that the identified proteins within the reproductive system could serve as avenues for the design of novel treatment options or drug targets. However, how the findings from this study can be implemented is unclear since the proteins are enriched in ciliated cells of the fallopian tubes. Still, proteins are also “expressed” in other tissues. – this needs to be included in the discussion.

We agree with the reviewer that this statement is not clearly motivated. This sentence has been removed in the revised version of the manuscript.

7. Information in Fig. S3B needs to be included in the main results section as the manuscript highlights its clinical translationability. The information on the overlap of proteins identified with the ciliopathies database should be included in the main figure.

We agree with the reviewer that this figure is important to highlight since it shows a link between identified proteins and disease. We have added a simplified version of this data in Figure 2D in the revised version.

8. Fig. S3B, CFAP45 showed a matching with three different types of ciliopathies. However, the manuscript did not mention or discuss the importance of this protein in reproductive function, i.e., do patients with the mutation in the CFAP45 gene have reproductive defects? This could also be functionally tested using a conditional knockout mouse model, specifically in the fallopian tubes.

We agree with the reviewer and have therefore added a section to the discussion in the revised manuscript, where we discuss the role of CFAP45 in reproductive function. Arranging a knockout mouse model experiment is outside the scope of this study due to limitations in time and resources. However, in a study by Dougherty et al (<https://doi.org/10.1038/s41467-020-19113-0>), CFAP45 was shown to be associated with reproductive defects, specifically asthenospermia due to impaired sperm flagellar motility. CFAP45-deficient individuals exhibit

dyskinetic sperm flagella, which compromise male fertility. This phenotype is also recapitulated in *Cfap45* knockout mice.

Reviewer #2 (Remarks to the Author):

This report takes a deep dive into the proteome of ciliated cells of the fallopian tube. Proper functioning of ciliated cells in tubal epithelia is essential for reproduction, and so a more comprehensive understanding of key structural and functional proteins in these cells is important for fertility research.

The manuscript is rooted in re-analyses of data from the Human Protein Atlas. The authors create a low-density TMA and stain for 130 proteins nominated, to understand whether the cilia-associated proteins are unique to the fallopian tube ciliated epithelium or are expressed by other ciliated cell types. The authors also mine published single cell expression data sets, a spatial single-cell proteomic dataset, and The CiliaCarta compendium to curate a comprehensive set of fallopian tube cilia associated proteins and annotate the subcellular localization of those proteins.

This report is clearly presented, easy to follow and well written. It is an impressive descriptive study but would benefit substantially from functional validation – for example, if any of these proteins are depleted in in vitro or in vivo models, does that impact cilia function or survival of ciliated epithelial cells? Insight into which (if any) of these proteins exhibit dysregulated expression in tubal pathologies would help prioritize which proteins are most important to study for human health and reproduction.

We thank the reviewer for the positive feedback and constructive comments regarding our manuscript. To add functional validation, we have included a fallopian tube sample from a patient with diagnosed hydrosalpinx in the revised version of the manuscript and performed IHC stainings of the least characterized proteins included in our study.

Minor point

- Figure S2, the authors should consider including expression of cell-type defining markers to support cell type assignments

The cell-type defining markers of the scRNAseq data clusters have been included in the revised version of the manuscript.

Reviewer #3 (Remarks to the Author):

This research aims to identify new proteins associated with cilia in the human Fallopian Tube (FT) by using a comprehensive approach that combines high-throughput transcriptomics (both bulk and in silico single-cell RNA sequencing) and proteomics (using immunohistochemistry).

The authors' recognition of the significance of expanding knowledge and investigating less explored areas is noteworthy. This study examines molecular differences (at the RNA and protein levels) in a vital organ for women's reproductive health—the fallopian tube—.

However, a notable concern arises regarding the utilization of in silico data and public datasets for selecting candidate genes to validate within the chosen sample set for the study. Specifically, relying on three single-cell RNA sequencing datasets to evaluate the cell specificity of the identified genes through bulk analyses may introduce bias and may not lead to robust conclusions. Consequently, the resulting study primarily provides descriptions and lacks functional reproducibility, as the data come from external sources rather than being internally generated.

We thank the reviewer for the positive feedback and constructive comments regarding our manuscript. The specific comments have been addressed below.

Introduction

1. While authors perform an extensive description of the motile cilia, it is only supported by few general references. Please, add literature supporting specific mechanisms such as the active movement of a motile cilium (page 3, lines 58-59), dynein arms (lines page 3 60-62).

We have added relevant references to the various parts of the introduction in the revised version of the manuscript.

2. Page 3, lines 68-69: The statement “The controlled import and export to and from cilia generates a unique protein repertoire in cilia” may suggest that such repertoire is well studied and established as a standard reference. However, it is crucial to provide clear interpretation and proper referencing on the extent of existing studies in this area, especially since the primary aim of the paper is to pinpoint and elucidate essential proteins associated with ciliary function.

We agree with the reviewer about the importance of proper descriptions and referencing. We have added relevant references to the indicated part of the introduction in the revised version of the manuscript.

3. Page 4, lines 87-88: please, ensure to include in this section the most relevant “other tissues with cells employing motile cilia” that were analysed in this study and contrasted with FT.

We agree with the reviewer and “Other tissues with motile cilia” have been specified.

Results

Overall, I find the results section well-written and comprehensive. However, I noticed a missing section detailing the experimental workflow employed in this study. It would greatly enhance the clarity and understanding of the paper if you could provide a concise description of the experimental workflow, outlining the sequence of steps and methodologies utilized in your research. Additionally, I would advise you to consider preparing a figure illustrating the experimental workflow. Such a visual aid would significantly improve the readability and comprehension of your manuscript for the reviewers and the readers.

We thank the reviewer for the proposal. To give a better overview of the experimental workflow, we have added an illustration to Figure 1 in the revised version.

The FT-elevated transcriptome

1. Page 6, Lines 103-108: the elucidation regarding the HPA project classification system used in this study should be included in the Methods section under “HPA RNA data summary”. Combining this information with the details provided in the Methods section concerning the gene categorization based on tissue type expression specificity (page 20, lines 479-485) would be beneficial. Consolidating these explanations within the same section would enhance clarity and coherence.

A text describing the rationale of the RNA specificity classification system has been added to the first section of the Results in the revised version of the manuscript.

2. If the creation of the FT-specific gene set involves its inclusion in the bulk analysis, what is the reasoning behind the decision behind excluding the ciliated cells from the FT in the HPA scRNA-seq dataset?

The previous version of the manuscript was based HPA data that did not include ciliated cells from FT. We have now updated the study to include data from a newer version of HPA with scRNA-seq data from ciliated cells from FT, endometrium, lung and bronchus. The implemented

changes include marginal changes to the number of FT-specific genes and associated subgroups, as well as changes to the analyses illustrated in Figure 1.

3. Page 7, lines 136-37: Could you please provide the data for which two genes had uncertain evidence?

We have clarified this in the Results section. Furthermore, the pie chart illustration of the numbers of each evidence level group has been removed from Figure 1, and the data is instead clearly presented in bar charts in Figure 2.

The spatial localization of FT-elevated proteins in ciliated and non-ciliated cells

1. Page 7, lines 149-150: please provide more explanation on what the reliability scores “Approved, Supported and Enhanced” mean.

An explanation of the reliability scores of the antibody-based data in the HPA has been added to the indicated part of the results section in the revised manuscript.

2. Page 8, line 183: please rephrase the sentence to clarify whether the total count of identified proteins is 29, with 7 being unique to the cilium base, or if out of 36 proteins, 29 are common and 7 are specific to the cilium base.

As part of a larger restructure of this part of the results section, the indicated sentence has been removed in the revised version of the manuscript.

3. Page 9, lines 191-197: Although illustrative for the proteins mentioned previously, this paragraph is only referring to model organisms. As mentioned by the authors there is a knowledge gap regarding the level of evidence we have for some proteins, but consider rephrase this paragraph, considering that transcripts and proteins are not always conserved between animal models and human.

As part of a larger restructure of this part of the results section, the indicated sentences have been removed in the revised version of the manuscript.

Spatial analysis of cilia-associated proteins in other tissues with motile cilia.

1. Page 11, line 240-249: in this paragraph the authors discuss the proteins detected across various analysed tissues. They provide insights into the known functions of some proteins (e.g. GAS2L2 or CRTAC1) while indicating if others are uncharacterized (e.g. KIAA2012). However, no information is provided for CFAP73, PNOC and SLC27A6. Please, confirm if these proteins are characterized to maintain consistency with the provided list.

As part of a larger restructure of this part of the results section, the indicated sentences have been removed in the revised version of the manuscript.

Validation of FTE cell specificity

1. Page 12, line 283: the Deep visual Proteomics (DVP) workflow is first mentioned here, however, no citations or explanations regarding this approach are provided in the Methods section. It is recommended to include comprehensive information about this algorithm in the Methods section to ensure clarity and understanding.

A comprehensive description of the DVP method and how it was implemented in this study was added to the method section in the revised version of the manuscript along with the associated reference.

2. Page 13, lines 301-307: Consider moving this paragraph to the discussion, since it does not provide any results per se.

The indicated paragraph has been removed and is instead exclusively discussed in the discussion section of the revised manuscript.

3. Page 13, lines 308-315: Considering the various tools and resources employed during this validation step, is there any evidence from in vivo studies or protein-protein interactions indicating the involvement of the examined proteins in ciliary function? The credibility of results, their interpretation, and extrapolation heavily rely on the level of evidence in such studies. I strongly suggest conducting a re-analysis where the level of evidence for each gene/protein is treated as a variable.

We agree with the reviewer about the importance of communicating the level of evidence for the proteins characterized in this study. In an effort to clearly convey the level of evidence together with our IHC localization data for each of the uncharacterized proteins, we have organized the 123 proteins that were spatially localized within ciliated cells into seven different groups according to whether or not there are existing data at the protein level, data at a cilia database and any GO term associated with the proteins. The groups are illustrated in a new Figure 3, where the proteins of each evidence group have been subdivided according to their annotated localization within ciliated FTE cells. IHC examples from each group are shown in Figure 4.

4. Page 13-14, lines 317, 329, 339: in these lines there are data presented as fractions of the total of proteins characterized: “a quarter of the proteins”, “two thirds of the 130 proteins”, “a sixth”, “an additional eight”. This is a strange way of presenting data; I suggest providing the precise numerical values instead.

This has been corrected in the revised version of the manuscript. The indicated type of data has been presented in numerical values.

Discussion

1. Page 15, lines 348-350: Concerning the subcellular localization of the proteins, emerging techniques such as spatial transcriptomics or proteomics are gaining traction, offering potential advancements in resolution and reproducibility at the single-cell or single-spot level. Please, provide the rationale for having chosen IHC over these novel techniques, especially considering their potential for enhanced resolution, reproducibility, and quantitative analysis. This rationale becomes even more crucial when studying the ciliated epithelium, a histological subtype that demands the utmost precision to achieve results at the subcellular level.

We agree with the reviewer about the current diversity of exciting emerging spatial techniques. As with all techniques, there are strengths and weaknesses. Antibody-based proteomics is to date, still the main method for determining spatial expression at a single cell or subcellular level due to high resolution, but at the price of less quantitative measurements. Through the HPA project, we have established a high-throughput workflow for IHC and antibody validation to produce spatial data with high reliability.

2. Page 10, lines 217-218: In accordance with the authors' findings, samples of the testis have been incorporated into the study to facilitate a comparison of expression and localization between cilia and flagellum. However, I find the discussion on this comparison somewhat lacking, particularly regarding the genes expressed in both types of structures, especially considering whether this tissue was included specifically for this purpose.

The topic of similarities and differences in protein repertoire between motile cilia and the flagellum has been more thoroughly discussed in the revised version of the manuscript.

3. Page 16, line 382, page 16: Please, express data as an exact number, not as a fraction (“a fourth of the 130 proteins”).

This has been corrected in the revised version of the manuscript. The indicated type of data has been presented in numerical values.

4. Page 18, lines 424-429: I suggest providing additional information about intercalary cells since the authors highlight their significance within the FTE. It would be interesting to include further data describing the protein expression on these cells even if it is not specific to them. Additionally, it would be valuable for the authors to propose potential reasons for the absence of proteins specifically localized to intercalary cells.

We agree that it would be interesting to further investigate the heterogeneous cell types of the FTE. However, since it is not possible to easily distinguish intercalary cells through classical IHC and none of the investigated proteins were exclusively expressed in intercalary cells, it was not possible to provide any reliable data on the expression profile of these cells. The absence of

intercalary-specific proteins may reflect their transitional phenotype, low abundance, or limited molecular distinction from neighboring cell types, which could challenge detection by both IHC and proteomic approaches. We have clarified this point in the revised.

5.Regarding the limitations of the study, it is noteworthy to mention that the sample size for fallopian tube tissue samples is relatively small, comprising only five samples, which may limit the generalizability and robustness of the findings. Apart from that, there is only three samples from endometrium, one sample from choroid plexus, and two samples for the rest of the tissues remaining (testis, caudate, bronchus, nasal mucosa and epididymis), that might not be sufficient to raise strong conclusions about the gene and protein expression in the studied subjects. Please, consider adding more samples to the sample size.

We thank the reviewer for the comment and have expanded the limitations of the study in the discussion in the revised version of the manuscript.

Methods

Tissues and ethical declaration

This section should address a more elaborated description and codification of both the donors and the samples.

1. Additional information regarding the cohort description (age, potential infections, surgical indication, menstrual cycle in the case of reproductive tissues) is required, as they could impact the features and function of the analysed tissues. It is also crucial to include details regarding the specific criteria used for inclusion and exclusion, as well as the methods utilized for determining these criteria. Please, provide a supplementary table clarifying these details while maintaining donor anonymity.

Regarding the acquisition of tissue specimens, please, provide details on the sample collection process, the age of the blocks, and any potential discrepancies in processing fresh tissue into formalin-fixed paraffin-embedded (FFPE).

All the tissues used in the study have been collected as described in the “Tissues and ethical declaration” in the Methods section. All tissue samples were anonymized in accordance with the approval and advisory report and no information regarding e.g. potential infections, surgical indication, menstrual cycle can be retrieved from surgical records.

2. Page 20, lines 470-475: What is the significance of the "y" character accompanying the number for each sample type? This is somewhat perplexing as it could be interpreted as either indicating the age of the FFPE block, the age of the tissue donor, or serving as an anonymized code for the sample. If it is meant to denote the anonymized code, it appears confusing that donor 25y has contributed both fallopian tube tissue and testis tissue. Please provide clarification on this matter and assign appropriate anonymized names to the samples.

This has been clarified in the Methods section in the revised version of the manuscript.

HPA RNA data summary

1. Page 20, line 478: As noted earlier, incorporating the explanation of the HPA project classification system outlined in the Results section (page 6: lines 103-108) could enhance comprehension and serve as an introductory guide to the categories detailed subsequently.

A text describing the rationale of the RNA specificity classification system has been added to the first section of the Results in the revised version of the manuscript.

2. It would be also helpful including a summary table that collects the categories discussed in the text along with their corresponding explanations. This will offer readers a concise representation of the different classifications thereby enhancing clarity and understanding of the content.

As specified previously, the RNA classification has been clarified in the revised version of the manuscript.

Single- cell RNA-seq data integration

1. Despite the integration of the three publicly available single-cell datasets is acknowledged, there is a lack of information regarding the specific integration among bulk RNAseq and single-cell RNAseq data. Please, add this information for clarity.

The integration of bulk and single-cell RNA data was not generated in this study, and the data were retrieved from version 23 of the Human Protein Atlas. This has been clarified in the revised version of the manuscript.

2. Page 21, lines 489-500: please, provide specific references to single-cell atlases or available datasets that were used when referring to “well-known protein markers of the fallopian tube cell type”.

We have removed the statement “well-known” in the revised version of the manuscript.

Annotation

1. Page 22, lines 529-530: how were the four levels of staining intensity determined? Was there a quantitative method employed for these determinations? Please, specify the image processing software used for managing the IHC results, along with the relevant parameters for image evaluation or the statistical analysis conducted.

We have added the IHC image analysis information to the annotation part of the methods section in the revised version of the manuscript.

2. Page 22, line 532: how was the distinction between “ciliated and non-ciliated cells” determined? Please indicate whether there was assessment conducted by a certified pathologist or if there was a double observer approach to confirm these findings.

We have detailed the annotation procedure in the indicated section in the revised version of the manuscript.

Data availability

-Page 24, line 574: It seems that the provided link www.v21.proteinatlas.org is not working correctly. Please, provide a valid link.

We have edited the indicated link in the revised manuscript.

Figures and legends:

Typically, figures should be comprehensible solely by referring to their accompanying legends. However, the current figure legends are insufficiently detailed to be self-explanatory. It is recommended to either enhance the clarity of the figures and to improve the comprehensiveness of the figure legends.

We have made changes to the figures and enhanced the comprehensiveness of the figure legends in the revised submission in order to better guide the readers.

-Figure 1B: This representation of a GO enrichment seems a bit confusing, and the smaller boxes do not have enough resolution to for effective reading. Exploring other visualization options for GO enrichment plots that incorporate statistical values such as p-values, z-scores, or defined thresholds would be beneficial in ensuring the enriched functions are more easily discernible and comprehensible.

We have supplemented the article with a Table S3 that lists the results from the GO enrichment analysis.

-Figure 1C: Similar to Figure 1B, I recommend considering alternative plots such as bar charts, stacked bar charts, or line graphs, which provide clear numeric references for each category and facilitate easier comparison between categories. Implementing such plots is likely to enhance the clarity and understandability of your data visualization.

We have edited the figures in the revised submission to enhance clarity, including moving Figure 1C to Figure 2, focusing on the 133 proteins investigated by IHC, where the GO term coverage of the 133 genes/proteins are illustrated in three bar charts.

-Figure Legend 1: The abbreviations within section (B) of the legend should be relocated to section (C) of the figure, as they pertain to that part of the illustration.

We have changed the figure and the abbreviations are no longer used in the figure in the revised version of the submission.

-Table S1: The category "Detected in fallopian only" in the table is labelled as "Detected in single" in the legend. Please ensure uniformity by using the same terminology in both the legend and the table.

The legend has been edited to match the table in the revised version of the submission.

-Figure 2A: This figure seems somewhat unclear; a more detailed explanation of the elements depicted in the graphics above and to the right, as well as clarification on the numerical values associated with them, would greatly enhance its clarity.

We have edited the figure, now labeled as Figure 2C, to enhance its clarity.

-Figure 2B: Please specify in the figure legend the type of tissue shown.

We have made sure to include tissue descriptions to IHC figures in the revised submission.

-Figure 3A: Within the intersection plots, there are two types of bars: (1) vertical bars with numerical values above them and (2) horizontal bars representing subcellular compartments, both accompanied by numerical annotations. Please indicate in the legend what specifically "numbers" indicate from the figure.

The plot has been simplified and the numbers have been clearly explained in the legend in the revised submission.

Minor comments:

1. Page 8, line 159: FTE, please specify acronyms when first mentioned.

We have specified this acronym when first mentioned in the revised version of the manuscript.

2. Page 8, line 167: please, replace the expression "well studied" by a most concise concept and consider incorporating levels of evidence (level of evidence in vivo / in silico, curated/ non-curated, etc...) where applicable.

We have rewritten this part and clarified evidence levels in the revised version of the manuscript.

3. Page 10, line 211-212: please provide the proper reference for this sentence.

We have added a reference to this sentence in the revised version of the manuscript.

4. Page 11, line 248: please specify of which tissue is OVGP1 a well-known marker.

This part of the results section has been rewritten in the revised version of the manuscript. As a result, OVGP1 is no longer mentioned as a well-known marker.

5. Page 12, line 278: please, separate “whereof” to “where of”

This part of the results section has been rewritten in the revised version of the manuscript. As a result, “whereof” was removed.

6. Page 14, lines 335-337: please provide the proper reference for these sentences

Since results regarding ciliopathy databases are moved to Figure 2, these sentences have been moved to the “The spatial localization of FT-elevated proteins in ciliated and non-ciliated cells”- part of the results section in the revised version of the manuscript. References have been provided to the indicated sentences.

7. Page 20, line 472: Given the coding system applied to all patients, it's possible that “male 37” might be a typo error and actually refers to “male 37y”, correct?

The age abbreviations have been clarified in the Method section in the revised version of the manuscript.

8. Page 21, line 514: IWGAV, please specify acronyms when first mentioned.

We have specified this acronym when first mentioned in the revised version of the manuscript.

Reviewer #4 (Remarks to the Author):

Review of “A high-resolution spatial map of cilia-associated proteins based on characterization of the human fallopian tube-specific proteome” by Hikmet et al.

Background and importance: Ciliated cells in the human body play a very important role in human physiology and disease. As such, the main aim of the authors to add to the collection of specific genes expressed in ciliated cells is important and laudable.

However, the execution of their plan leaves much to be desired, to the extent that I cannot recommend publication, at least not in anything like the current forms.

What the authors did: This paper mines data from the Human Proteome Atlas, a very useful community resource that the authors are leading or part of. Despite the name, it contains much RNA expression data, from which the authors start to identify genes with elevated expression in the fallopian tube compared to other tissues. They found 315 genes categorized as "tissue-enriched", "group-enriched", or "tissue-enhanced" in the fallopian tube. Gene Ontology analysis confirms that a majority of these genes are associated with motile cilia function and sperm/flagellar motility. Out of the 315 genes, the authors then selected 130 genes with reliable antibodies available from the Human Protein Atlas for further immunohistochemistry (IHC) profiling. If I understand correctly (not very clear in the manuscript), the sole experimental work of the paper is then the in-depth IHC analysis on a tissue microarray containing fallopian tube and other tissues with motile ciliated cells (e.g., respiratory tract, brain ventricles). The purpose is to map the subcellular localization of the 130 proteins, identifying their association with ciliated cells, non-ciliated cells, or specific ciliary compartments. The authors then consult three sources of single cell information to validate their IHC findings using: a. Single-cell RNA-seq data from fallopian tube epithelial cells b. Mass spectrometry data from laser-captured ciliated (FOXJ1+) and non-ciliated (FOXJ1-) fallopian tube cells and c. Comparison with existing cilia databases (CiliaCarta, SysCilia, etc.).

An obvious critique of this effort is that it is entirely descriptive: No functional experiments are performed and no new biological insights are forthcoming. That by being the case, the authors would have had to have a very solid resource outcome, i.e. one would have expected a golden set of proteins that the cilia community can now study. Unfortunately, this is not the case.

The initial start from bulk RNAseq is questionable as this involves at least two major epithelial cell types that are mixed (plus some others), making it difficult to do the enrichment analysis. The authors should come up with a more inclusive list of proteins from more of their sources (and the recent single cell atlases?). They should then devise a logical scheme of how to reduce this catalogue to a smaller set of high confidence proteins that fulfill some combination of evidence. Maybe a score with some rational basis instead of a purely subjective discussion, ideally even some rudimentary machine learning based on the multiple data sets. In any case some elementary statistics is necessary. Some functional validation would of course add tremendously to the paper although the reviewer realizes that this is easier said than done.

The novel proteins should be discussed as for their role in ciliated cells. It should be made clear what was known before and what is novel after this paper. In general, while the authors provide some biological interpretation, they could further elaborate on the potential functional roles and mechanisms of the novel cilia-related proteins they identified, based on literature and bioinformatics analyses. They should explore the potential implications of their findings for understanding ciliopathies, infertility, and other disorders related to cilia dysfunction, and discuss how their data could inform future studies in these areas. The authors could discuss the limitations of their study, such as the potential for missing low-abundance proteins or proteins with poor antibody performance, and how these limitations could be addressed in future work.

We thank the reviewer for the feedback and constructive comments regarding our manuscript. The specific comments have been addressed below.

Specific comments

1. The IHC images shown in this manuscript are a valuable resource for researchers interested in the spatial localization of FT-related proteins. Due to figure limitations, not all staining images can be shown in the main publication. While the IHC images will be available at the Biostudies repository, an easy-accessible database or extended supplementary file may be helpful for the community.

As contributors of the open-access HPA resource, we strongly agree with the reviewer about the benefit of providing easy access to generated scientific data for the research community. We have made the stainings available through the Biostudies repo, and the stainings can be downloaded one by one.

2. In Figure 2B and the corresponding text, a localization of proteins mainly to the nucleus of ciliated FTE cells is described. Visually, it is correct that the proteins are located in the nucleus by staining, but they also seem to be visible in the cytoplasm and cilia (e.g. CAPS). In contrast, FOXJ1 is exclusively located in the nucleus. EFCAB1 seems to be also positive in the nucleus, but is associated to the group of cytoplasmatic markers. The distribution should be sub-grouped (e.g. exclusive nuclear, nuclear + cytoplasm etc.). Also sentences such as 'localization to mainly ciliated FTE cell nuclei' could be adapted accordingly.

The results section in the revised version of the manuscript has been adapted to focus on previously uncharacterized proteins that were spatially localized in this study. As a result, figures were remade and groups of genes/proteins with the same subcellular localization are no longer shown as grouped IHC images. The complete combination of subcellular localization was clearly stated in the text of the revised manuscript, where localization was mentioned.

3. Line 245: After discussing the results of Figure 3B, the authors come back to results displayed in Figure 3A. A reference to Fig. 3A is missing here and the authors may consider re-arranging this section.

Since the results section in the revised version of the manuscript has been adapted to focus on previously uncharacterized proteins that were spatially localized in this study, figures have been remade with IHC images of other genes/proteins, and the results section has been rewritten accordingly. The indicated part of the text has therefore been removed. The rewritten parts were made with the reviewer's point in mind.

4. Line 251: The IHC result of SLC27A6 is described and referred to Figure 3B. However, the staining image is not displayed here.

As mentioned above, this part has been rewritten, and the indicated part has been removed.

5. Ciliated cells from FT were not included in the HPA scRNA-seq dataset, but the authors re-visit datasets from Hu et al., Dinh et al. and Ulrich et al. There is another single-cell resolved atlas on the post-menopausal fallopian tube by Lengyel et al. (PMID: 36543131) published in 2022. It would be interesting to see this dataset integrated as well and to elaborate on the pre- and post-menopausal state of the fallopian tube for the most relevant proteins of your dataset. This could be of relevance as Hu et al. describes post-menopausal FT, while Dinh et al. shows all (but one) premenopausal FT.

We agree with the reviewer about the importance of studying menopausal changes in protein expression in the FT. However, our current study focuses primarily on validating protein-level expression patterns using IHC and spatial proteomics, rather than performing a comprehensive transcriptomic meta-analysis. Therefore, we have added this as a future perspective in the Discussion.

6. Figure 4A: Based on the integration of the three scRNA-seq datasets and mapping the resulting clustering on your FT-specific proteins, the authors may discuss in more detail the difference between the ciliated 1 vs ciliated 2 population. Similarity to secretory cells is indicated, but is there a biological translation of the findings? If the expression signal was consistently higher in the population of ciliated 1 cells, could this be a systematic technical effect?

To simplify the comparison with IHC data, different scRNAseq clusters of the same cell type have been merged in the revised version of the manuscript.

7. In Figure S3B, the authors depict ciliopathy-related genes and conclude that a sixth of the 130 genes is associated to misfunctions of the cilia. It would be interesting to discuss examples and how this dataset could contribute to suggest to targets to treat or understand ciliopathies.

We have added a discussion regarding the spatially analyzed proteins and ciliopathy.

8. Infertility is mentioned as a main motivator to study the fallopian tube. The readership would profit from a more detailed analysis and discussion on fertility-associated genes and how this study is contributing to understand the pathways disentangling pathways involved in infertility as written in the abstract.

An analysis of FT tissue samples from women with hydrosalpinx has been added to the study, accompanied by a discussion regarding the analyzed proteins in this disease sample.

9. Apart from the above comments, the figures are very nicely and professionally done.

We thank the reviewer for the positive comments regarding the design of the figures.

Minor comments

1. Figure 1C: evidence and GO term level are difficult to separate in figure. A visual clarification could help.

We have supplemented the article with a Table S3 that lists the results from the GO enrichment analysis.

2. Line 155 to line 158: “proteins made up the largest proportion of the 130 selected proteins. Is this referring to protein subunits and isoforms or rather quantitative level/abundance? Please explain the term “proportion” in more detail.

Since the results section in the revised version of the manuscript has been rewritten, this sentence has been removed.

3. Figure 2B, Figure 3A: (representative) scale bars of microscopy images are missing.

Scale has been defined in the figure legends in the revised manuscript to clarify the size of the tissue images.

4. Figure 4B: the color of the grey grid lines interferes with the readability of database labels. Opaque grid lines could improve the readability of the figure content.

The plot has been remade slightly to increase the clarity of the database texts and moved to supplementary Figure 4. Also, a new variant of the circle plot has been made as an addition to Figure 2, where databases are merged to create a simplified view of database inclusion for the 133 proteins analyzed with IHC.

5. Line 378: ‘non-FOXJ1-positive’ - rephrase to FOXJ1 negative

The indicated sentence has been adjusted in the revised version of the manuscript.

RESPONSE TO REVIEWERS

Reviewer #1

1) Please clarify whether the regions obtained for figures 7C-D and S8 were derived from the fimbria for all samples, including controls and hydrosalpinx.

All samples in the additional analysis are from the ampulla region. This has now been clarified in the method section and the figure legend.

2) Please acknowledge in the discussion that the hydrosalpinx sample was derived from an n of 1; therefore, it is challenging to draw accurate conclusions from this finding. This reviewer understands the difficulties in obtaining hydrosalpinx samples. However, this caveat must be presented to the readers.

A section on the limitations of the conclusions as only one hydrosalpinx sample was used has now been added to the discussion.